# Fast bacterial growth reduces antibiotic accumulation and efficacy

Urszula Łapińska[1,2]*, Margaritis Voliotis[1,3], Ka Kiu Lee[1,2], Adrian Campey[1,2], M Rhia L Stone[4,5], Brandon Tuck[1,2], Wanida Phetsang[4], Bing Zhang[4], Krasimira Tsaneva-Atanasova[1,3,6,7], Mark AT Blaskovich[4], Stefano Pagliara[1,2]*

[1]Living Systems Institute, University of Exeter, Exeter, United Kingdom; [2]Biosciences, University of Exeter, Exeter, United Kingdom; [3]Department of Mathematics, University of Exeter, Exeter, United Kingdom; [4]Centre for Superbug Solutions, Institute for Molecular Bioscience, The University of Queensland, Brisbane, Australia; [5]Department of Chemistry and Chemical Biology, Rutgers, the State University of New Jersey, Piscataway, United States; [6]EPSRC Hub for Quantitative Modelling in Healthcare, University of Exeter, Exeter, United Kingdom; [7]Department of Bioinformatics and Mathematical Modelling, Institute of Biophysics and Biomedical Engineering, Bulgarian Academy of Sciences, Sofia, Bulgaria

**\*For correspondence:**
U.Lapinska@exeter.ac.uk (UŁ);
s.pagliara@exeter.ac.uk (SP)

**Competing interest:** The authors declare that no competing interests exist.

**Abstract** Phenotypic variations between individual microbial cells play a key role in the resistance of microbial pathogens to pharmacotherapies. Nevertheless, little is known about cell individuality in antibiotic accumulation. Here, we hypothesise that phenotypic diversification can be driven by fundamental cell-to-cell differences in drug transport rates. To test this hypothesis, we employed microfluidics-based single-cell microscopy, libraries of fluorescent antibiotic probes and mathematical modelling. This approach allowed us to rapidly identify phenotypic variants that avoid antibiotic accumulation within populations of *Escherichia coli*, *Pseudomonas aeruginosa*, *Burkholderia cenocepacia,* and *Staphylococcus aureus*. Crucially, we found that fast growing phenotypic variants avoid macrolide accumulation and survive treatment without genetic mutations. These findings are in contrast with the current consensus that cellular dormancy and slow metabolism underlie bacterial survival to antibiotics. Our results also show that fast growing variants display significantly higher expression of ribosomal promoters before drug treatment compared to slow growing variants. Drug-free active ribosomes facilitate essential cellular processes in these fast-growing variants, including efflux that can reduce macrolide accumulation. We used this new knowledge to eradicate variants that displayed low antibiotic accumulation through the chemical manipulation of their outer membrane inspiring new avenues to overcome current antibiotic treatment failures.

## Editor's evaluation

This study addresses mechanisms by which bacteria are able to survive and evade killing by antibiotics. Using fluorescent versions of antibiotics it studies whether entry/efflux of the drug itself is a significant contributor to the observed variability of antibiotic activity. This study will be of interest to microbiologists and clinicians for the design of better antibiotic therapies and improves our understanding of the relationships between drug uptake, bacterial growth, and drug efficacy.

## Introduction

Phenotypic heterogeneity between genetically identical cells has been observed across all three domains of life (*Richards et al., 2019*; *Ackermann, 2015*). This heterogeneity is characterised by

**eLife digest** Bacteria can cause an array of diseases ranging from mildly inconvenient to deadly. In fact, every year around the world, five million people succumb to a bacterial infection. Antibiotics can kill bacteria or stop their growth, but many bacterial species are now able to evade these drugs.

To be efficient, most antibiotics first need to get inside a bacterium; there, they accumulate until they reach the concentration they need to act. Often, the drugs make their way through channel-like structures ('pores') studded through the external membranes of bacteria and which control the passage of molecules in and out of cells.

Resistance usually emerges when genetic changes provide the microorganism with an advantage against antibiotics, or when the microorganism performs the biochemical reactions necessary for life at a slower pace.

In contrast, Łapińska, Pagliara et al. decided to examine how genetically similar *Escherichia coli* bacteria which differed in their growth rate would fare against antibiotics. The drug targeted ribosomes, the machinery that produces proteins in a cell. A combination of techniques was used to follow individual cells, revealing that fast-growing variants better managed to survive. A closer look showed that bacteria which were growing quickly had a surplus of ribosomes, which then produced more pores that could pump the antibiotic out the cell. Next, Łapińska, Pagliara et al. exposed the bacteria to both the antibiotic and a compound that weakens bacterial membrane; this erased the advantage shown by the fast-growing variants. Overall, this work gives a finer understanding of the mechanisms that underlie antibiotic resistance, which could help pave the way to new strategies to combat harmful bacteria.

individual cells that display differing phenotypic traits (*Golding et al., 2005*; *Lidstrom and Konopka, 2010*) and permit genotypes to persist in fluctuating environments (*Ackermann, 2015*). Phenotypic heterogeneity in the bacterial response to antibiotics contributes to antimicrobial resistance (*Windels et al., 2019b*; *Levin-Reisman et al., 2019*; *Brauner et al., 2016*; *Bamford et al., 2017*; *Goode et al., 2021a*; *Goormaghtigh and Van Melderen, 2019*; *Goode et al., 2021b*) and the failure to effectively treat bacterial infections (*Mulcahy et al., 2010*; *Helaine et al., 2014*; *Stapels et al., 2018*). Therefore, it is imperative to develop new diagnostics capable of rapidly identifying phenotypic variants that survive antibiotic treatment (*Baltekin et al., 2017*) and develop new antibiotic therapies against such phenotypic variants (*Shatalin et al., 2021*).

Here we hypothesise that this phenotypic diversification is driven by fundamental cell-to-cell differences in membrane transport mechanisms and their underpinning regulatory networks. In order for an antibiotic to be effective, it needs to reach its cellular target at a concentration that is inhibitory for micro-organism growth (*Rybenkov et al., 2021*). In gram-negative bacteria, intracellular antibiotic accumulation (*Rybenkov et al., 2021*; *Van Bambeke et al., 2006*; *Six et al., 2018*) is a complex biophysical phenomenon involving different physicochemical pathways and a combination of exquisitely regulated active and passive transport processes (*Rybenkov et al., 2021*; *Zgurskaya et al., 2018*). These processes include diffusion through the outer membrane lipid bilayer (*Rybenkov et al., 2021*) and porins (*Pagès et al., 2008*; *Nestorovich et al., 2002*); self-promoted uptake through the outer membrane (*Farmer et al., 1992*); diffusion through the inner membrane lipid bilayer which displays orthogonal selection properties compared to the outer membrane (*Silver, 2016*; *Cama et al., 2019*); active transport via inner membrane transporters (*Silver, 2016*); efflux out of the cell (*Du et al., 2014*; *Blair et al., 2016*; *Blair et al., 2014*; *Fitzpatrick et al., 2017*); enzymatic modification or degradation (*Rybenkov et al., 2021*); and eventually binding to the intracellular target.

Learning the rules that permit antibiotics to accumulate in gram-negative bacteria is vitally important in order to combat phenotypic and genotypic resistance to antibiotics (*Silver, 2016*; *Acosta-Gutiérrez et al., 2018*; *Tommasi et al., 2015*). However, most permeability data are sequestered in proprietary databases (*Rybenkov et al., 2021*). Moreover, such experimental datasets have often been generated via cell-free methods that permit the measurement of the diffusion rate of a compound through simplified membrane pathways (*Delcour, 2013*), but care should be taken when projecting these data to the more complex accumulation dynamics in live cells (*Rybenkov et al., 2021*). Live or fixed cell methodologies including radiometric, fluorometric or biochemical assays (*Kojima and Nikaido, 2013*;

*Piddock et al., 1999*; *Asuquo and Piddock, 1993*), mass spectrometry (*Zhou et al., 2015*; *Richter et al., 2017*; *Davis et al., 2014*; *Prochnow et al., 2019*; *Brochado et al., 2018*; *Iyer et al., 2018*; *Tian et al., 2017*), Raman spectroscopy (*Heidari-Torkabadi et al., 2015*), microspectroscopy (*Vergalli et al., 2018*; *Vergalli et al., 2017*; *Vergalli et al., 2020*), and fluorescence microscopy (*Reuter et al., 2020*) have also been employed to carry out antibiotic accumulation assays. These techniques generally rely on ensemble measurements that average the results obtained from a large population of micro-organisms, or are derived from examining only a handful of individual bacteria. Therefore, little is known about the variability in individual drug accumulation across many single cells within a clonal population.

Here, we fill this fundamental gap in our knowledge by harnessing the power of microfluidics-microscopy (*Łapińska et al., 2019*; *Cama et al., 2020*) combined with fluorescent antibiotic-derived probes (*Stone et al., 2018*; *Lin et al., 2021*; *Blaskovich et al., 2019*) as well as unlabelled antibiotics. This approach allows us to examine the interactions between the major classes of antibiotics and hundreds of live individual bacteria in real-time whilst they are being dosed with the drugs. Combined with mathematical modelling these data allow us to rapidly identify phenotypic variants that avoid antibiotic accumulation and are able to sustain growth in the presence of drugs without acquiring genetic mutations. We show that bacteria close to the antibiotic source accumulate faster membrane-targeting antibiotics but more slowly antibiotics with intracellular targets compared to bacteria further away from the antibiotic source. In contrast with the current consensus that slow cell growth leads to reduced antibiotic efficacy, we discover that fast growing phenotypic variants avoid macrolide accumulation due to a higher abundance of both ribosomes (i.e. the drug target) and efflux pumps. We further demonstrate that chemically manipulating the bacterial outer membrane permits us to eradicate variants that display low antibiotic accumulation. Adopting our novel approach in clinical settings to inform the design of improved drug therapies could aid refining our one health approach to antimicrobial resistance.

## Results

### Experimental assessment of single-cell real-time drug accumulation dynamics

We combined our recently developed single-cell microfluidics-microscopy platform (*Łapińska et al., 2019*; *Cama et al., 2020*; *Stone et al., 2020*) with a library of fluorescent derivatives representing most major classes of antibiotics, including macrolides (roxithromycin) (*Stone et al., 2020*), oxazolidinones (linezolid) (*Phetsang et al., 2014*), glycopeptides (vancomycin) (*Blaskovich et al., 2019*), fluoroquinolones (ciprofloxacin) (*Stone et al., 2019*), antifolates (trimethoprim) (*Phetsang et al., 2016*), and membrane-targeting lipopeptides/peptides (polymyxin B, octapeptin, and tachyplesin) (*Blaskovich et al., 2019*; *Figure 1A*).

Each antibiotic was functionalised at a site that minimises any changes in biological activity, adding a substituent that allows for facile coupling with a small fluorophore, nitrobenzoxadiazole (NBD, *Appendix 1—table 1*) as previously reported (*Blaskovich et al., 2019*; *Stone et al., 2020*; *Phetsang et al., 2014*; *Stone et al., 2019*; *Phetsang et al., 2016*). Using minimum inhibitory concentration (MIC) assays we found that the fluorescent derivatives of polymyxin B, octapeptin, tachyplesin, vancomycin, and linezolid maintained the antibiotic activity of the parent drug against *E. coli*, whereas the fluorescent derivatives of roxithromycin, trimethoprim, and ciprofloxacin displayed a 3-fold, 64-fold, and 256-fold increase compared to the parent drug, respectively (*Appendix 1—table 1*). Next we used each probe in our microfluidics-microscopy platform (*Łapińska et al., 2019*; *Cama et al., 2020*; *Stone et al., 2020*; *Glover et al., 2022*) to quantify the dynamics of the accumulation of each antibiotic in individual bacteria in real-time (*Figure 1B* and *Figure 1—source data 1*). Briefly, we loaded an aliquot of a stationary phase clonal bacterial culture in a microfluidic device equipped with small parallel channels, each hosting between one and six bacteria (*Łapińska et al., 2019*; *Cama et al., 2020*; *Stone et al., 2020*). Then we continuously flowed lysogeny broth (LB) medium into xthe device for 2 hr to stimulate cell growth and reproduction. During this period bacteria displayed an average elongation rate of (5.3±1.2) μm hr⁻¹. We also performed separate experiments flowing LB for a longer period of time and found that the elongation rate (averaged across the bacterial population) did not further increase after the first 2 hr exposure to LB (*Figure 1—figure supplement 1* and *Figure*

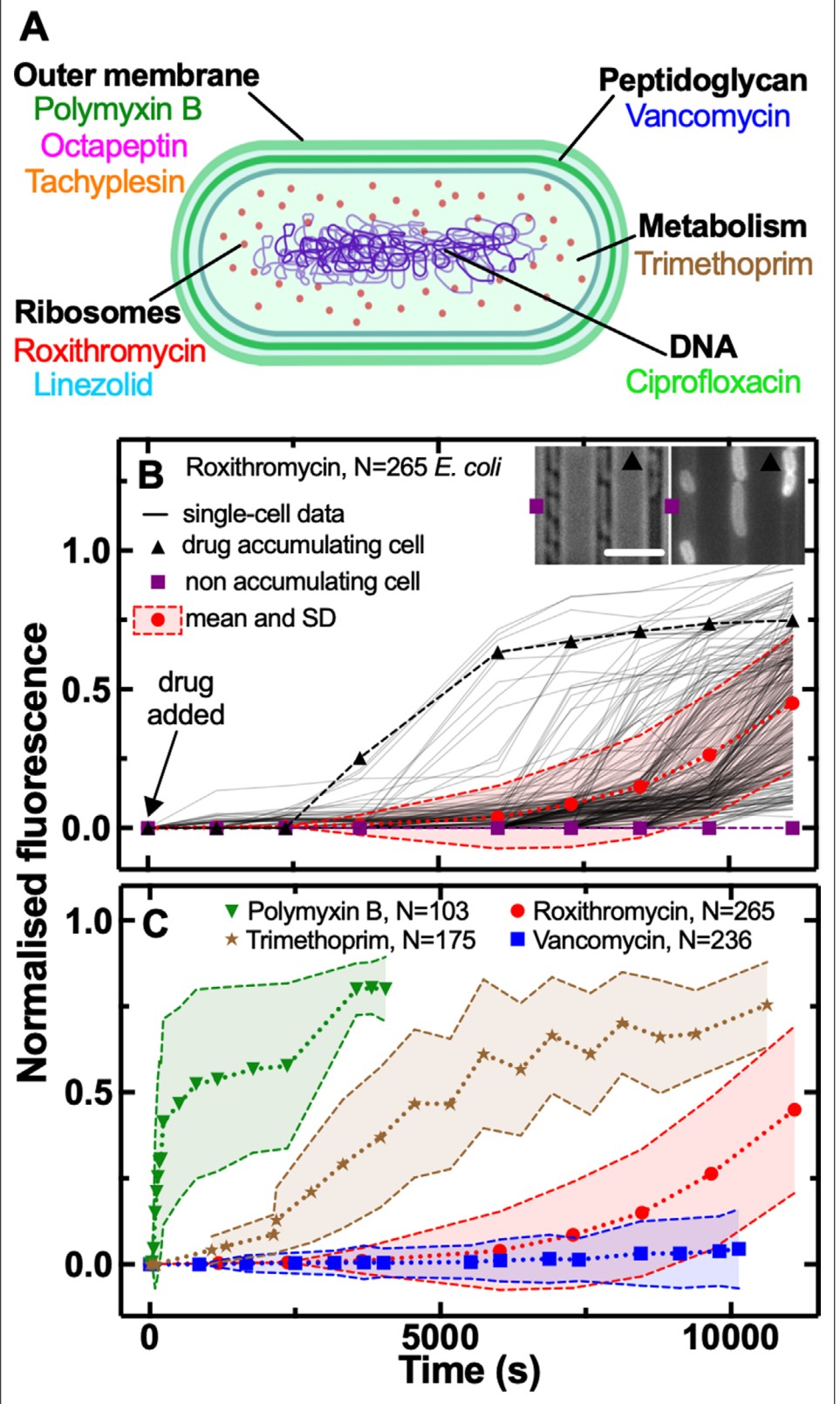

**Figure 1.** Phenotypic heterogeneity in the accumulation of the major classes of antibiotics. (**A**) Illustration depicting the eight antibiotics employed in this study alongside their bacterial targets. (**B**) Accumulation of the fluorescent derivative of roxithromycin in 265 individual *E. coli* (continuous lines) after adding the probe at 46 µg mL⁻¹ extracellular concentration in M9 minimal medium from t=0 onwards. Fluorescence values were background

*Figure 1 continued on next page*

*Figure 1 continued*

subtracted and normalised first by cell size and then to the maximum value in the dataset (see Methods). The circles and shaded areas represent the mean and SD of the values from 265 bacteria collated from biological triplicate. The squares represent the fluorescent values of a representative bacterium that does not accumulate the fluorescent derivative of roxithromycin, whereas the triangles represent the fluorescent values of a representative bacterium that accumulates the drug. Insets: representative brightfield and fluorescence images after 7000 s incubation in the fluorescent derivative of roxithromycin, the symbols indicate the two representative bacteria above. Scale bar: 5 μm. (**C**) Population average (symbols) and SD (shaded areas) of the accumulation of the fluorescent derivatives of polymyxin B (triangles), trimethoprim (stars), roxithromycin (circles), and vancomycin (squares) probes added at 46 μg mL$^{-1}$ extracellular concentration in M9 minimal medium from t=0 onwards. Data are obtained by averaging at least one hundred single-cell values (i.e. N=103, 175, 265, and 236, respectively) collated from biological triplicate. Corresponding single-cell data along with data for the fluorescent derivatives of linezolid, tachyplesin, octapeptin, and ciprofloxacin probes are reported in *Figure 1—figure supplement 2*.

The online version of this article includes the following source data and figure supplement(s) for figure 1:

**Source data 1.** Measurements of single-cell drug accumulation, size, elongation rate and doubling time in E. coli, S. aureus, P. aeruginosa and B. cenocepacia.

**Figure supplement 1.** Measurement of single-cell elongation rates.

**Figure supplement 2.** Measurement of single-cell drug accumulation.

**Figure supplement 3.** Measurement of normalised single-cell drug accumulation.

**Figure supplement 4.** Heterogeneity in the accumulation of different antibiotics.

**Figure supplement 5.** Measurement of single-cell doubling times.

**Figure supplement 6.** Interdependence between cell size and drug accumulation.

**Figure supplement 7.** Staining of bacteria with different antibiotics.

**Figure supplement 8.** Comparison of single-cell roxithromycin accumulation in *E. coli* and *S. aureus*.

**Figure supplement 9.** Comparison of single-cell vancomycin accumulation in *E. coli* and *S. aureus*.

**Figure supplement 10.** Comparison of single-cell ciprofloxacin accumulation in *E. coli*, *P. aeruginosa,* and *B. cenocepacia*.

**Figure supplement 11.** Impact of drug milieu and concentration on drug accumulation.

**Figure supplement 12.** Impact of drug labelling on drug accumulation.

**Figure supplement 13.** Main single-cell kinetic parameters inferred using our mathematical model.

**Figure supplement 14.** Second order single-cell kinetic parameters inferred using our mathematical model.

**Figure supplement 15.** Coupling between accumulation parameters.

*1—source data 1*), corroborating previous bulk data showing that *E. coli* is in exponential phase between 2 and 5 hr after passage in fresh medium (*Smith et al., 2018*). Therefore after 2 hr incubation in LB, we injected one of the antibiotic probes and imaged the real-time intracellular probe accumulation in hundreds of individual live bacteria (*Appendix 1—Videos 1 and 2*). Typically, upon onset drug accumulation increased until reaching steady-state saturation levels (*Figure 1B*) due to probe efflux, compound transformation (*Rybenkov et al., 2021*), or target saturation (*Silver, 2016*), although several bacteria displayed divergent accumulation dynamics (*Figure 1—figure supplements 2 and 3* and *Figure 1—source data 1*).

## Heterogeneity in antibiotic accumulation in gram-negative and gram-positive bacteria

These single-cell measurements revealed hitherto unrecognised phenotypic heterogeneity in intracellular drug accumulation in clonal populations of *E. coli* as evident from the microscopy images in *Figure 1B* and *Figure 1—figure supplement 2*. In contrast, standard techniques measure population averages of drug accumulation across thousands or millions of cells (*Six et al., 2018*; *Kojima and Nikaido, 2013*; *Asuquo and Piddock, 1993*; *Zhou et al., 2015*; *Richter et al., 2017*; *Davis et al., 2014*; *Prochnow et al., 2019*; *Tian et al., 2017*). In our single-cell assay, population averages (circles in *Figure 1B*) did not reflect the fact that some phenotypic variants displayed a remarkably delayed onset, slower uptake rate or reduced saturation with respect to other cells (e.g. compare the accumulation trajectories reported by the squares - no accumulation - vs triangles - high accumulation

- in *Figure 1B*). These phenotypic variants have thus far remained unrecognised in population-based experiments and give rise to large coefficients of variation (CV, the ratio of the SD over the mean) in the accumulation of each of the eight investigated antibiotics (*Figure 1C* and *Figure 1—figure supplement 4*). In the following, we will, therefore, use CV as a reporter for phenotypic heterogeneity within bacterial populations as previously reported (*Silander et al., 2012*).

It is worth noting that all bacteria within each experiment were exposed to the same concentration of probe (46 μg mL$^{-1}$) for the same duration and to the same drug milieu, i.e., minimal medium M9 to avoid dilution of probes due to cell growth (*Rybenkov et al., 2021*). As a consequence, during drug treatment bacterial divisions were rare events. For example, during treatment with roxithromycin dissolved in M9, none of the analysed bacteria underwent a full cell cycle from birth to division. Moreover, 15% of the bacteria analysed underwent one division during treatment but were born before drug treatment commenced with an average doubling time of (75±28) min (*Figure 1—figure supplement 5* and *Figure 1—source data 1*). In comparison, during treatment with roxithromycin dissolved in LB, 41% of the analysed bacteria underwent a full cell cycle from birth to division. Furthermore, when including bacteria that underwent one division during treatment but were born before drug treatment, the average doubling time (29±9 min) was significantly shorter compared to that measured for treatment in M9 (****, *Figure 1—figure supplement 5*). Moreover, in accordance with previous studies about phenotypic responses to antimicrobials (*Windels et al., 2019a*; *Pu et al., 2016*; *Attrill et al., 2021*), we found that bacterial variants displaying delayed or reduced antibiotic accumulation were genuine phenotypic variants, since DNA sequencing of the device outflow did not reveal any genetic mutations compared to untreated bacteria. Furthermore, these variants did not display significant differences in cell size (*Figure 1—figure supplement 6* and *Figure 1—source data 1*) and we further normalised each single-cell fluorescence value to the corresponding single-cell size (see Methods) (*Taniguchi et al., 2010*).

Due to the presence of these phenotypic variants, not all the bacteria were stained by each antibiotic probe, thus we found drug-dependent dynamics in the fraction of stained bacteria (*Figure 1—figure supplement 7*). The lipopeptide/peptide probes targeting the outer bacterial membrane (polymyxin B, octapeptin, and tachyplesin) stained 90% of the investigated bacteria within 1000 s post-addition to the microfluidic device. At this time, the trimethoprim and ciprofloxacin probes targeting intracellular components had stained only 50% of the bacteria, whereas the number of bacteria stained by roxithromycin and vancomycin probes, with a large molecular weight (1064 and 1650 g mol$^{-1}$, respectively), was close to zero. However, the roxithromycin probe did stain 50 and 90% of the bacteria around 7500 s and 9000 s, respectively, post-addition to the device, by which time only 15% of the bacteria had been stained by vancomycin. The lack of vancomycin staining was expected since vancomycin cannot cross the gram-negative double membrane to access its peptido-glycan target (*Murray, 1995*).

Next, we verified that this hitherto unrecognised heterogeneity in antibiotic accumulation is not a phenotypic feature exclusive to *E. coli*. When we compared and contrasted roxithromycin-NBD accumulation in *E. coli* against uptake in the gram-positive bacterium *S. aureus*, we found that although the latter displayed more rapid accumulation dynamics (*Figure 1—figure supplement 8* and ; *Figure 1—source data 1*), also *S. aureus* displayed phenotypic variants with delayed or reduced accumulation. In fact, roxithromycin-NBD reached saturation levels 3000 s post-addition in some *S. aureus* cells, whereas other bacteria accumulated the drug at very low levels and only by 5000 s post-addition (with a CV in range 53–372% and 29–73% for *E. coli* and *S. aureus*, respectively). In contrast, the gram-positive targeting vancomycin-NBD readily and homogeneously accumulated in *S. aureus* within 2500 s post-addition (CV in range 12–14%), but did not accumulate in *E. coli* (within this same timeframe, *Figure 1—figure supplement 9* and *Figure 1—source data 1*). Finally, we found phenotypic variants with delayed or reduced accumulation of ciprofloxacin-NBD in three clinically-relevant gram-negative bacteria: *E. coli*, *P. aeruginosa,* and *B. cenocepacia* (CV in range 12–329%, 24–534%, and 31–90%, *Figure 1—figure supplement 10* and *Figure 1—source data 1*). Furthermore, ciprofloxacin-NBD accumulated more slowly and to a lower extent in *P. aeruginosa* compared to *E. coli* and *B. cenocepacia* (*Figure 1—figure supplement 10*) in accordance with previous measurements at the whole population level (*Asuquo and Piddock, 1993*) and possibly due to the high porin impermeability in *P. aeruginosa* (*Ude et al., 2021*).

In order to verify that dilution of the intracellular concentration via bacterial doubling did not play a key role in the observed heterogeneity in antibiotic accumulation, we run separate controls using *E. coli* and roxithromycin-NBD dissolved either in M9 or LB. We found higher roxithromycin-NBD accumulation (*Figure 1—figure supplement 11* and *Figure 1—source data 1*) as well as shorter doubling times (*Figure 1—figure supplement 5*) when LB was used as drug milieu. We also found similarly large heterogeneity when roxithromycin-NBD was dissolved in M9 or LB (CV in range of 84–372% and 51–428%, respectively). These data demonstrate that heterogeneity in roxithromycin accumulation cannot be explained via dilution due to bacterial doubling. Finally, in order to verify that neither the drug concentration nor the labelling underpin the observed heterogeneity in antibiotic accumulation, we run separate controls using *E. coli* and different concentrations of roxithromycin-NBD and polymyxin B-NBD (*Figure 1—figure supplement 11* and *Figure 1—source data 1*), as well as unlabelled ciprofloxacin, ciprofloxacin-NBD, roxithromycin-NBD, and roxithromycin-DMACA (dimethylaminocoumarin-4-acetate, *Figure 1—figure supplement 12* and *Figure 1—source data 1*). When using the same excitation conditions used for our fluorescent derivatives (0.03 s exposure to the blue excitation band of a broad-spectrum LED operated at 8 mW, see Methods), unlabelled ciprofloxacin autofluorescence detected from the bacteria was not distinguishable from the background (i.e. from empty channels). However, unlabelled ciprofloxacin was distinguishable from the background upon 0.1 s exposure to the UV excitation band of a broad-spectrum LED operated at 40 mW. In all cases we identified phenotypic variants with delayed or reduced antibiotic accumulation, leading to large CVs as shown in *Figure 1—figure supplements 11 and 12*. We can also exclude possible effects of variations in magnesium availability (*Farmer et al., 1992*; *Peterson et al., 1987*) on the measured heterogeneity in antibiotic accumulation since all bacteria were exposed to the same medium within the microfluidic device.

## Single-cell coupling between kinetic accumulation parameters

Prompted by these novel findings, we moved on to an in-depth examination of antibiotic accumulation dynamics and the underlying cellular and molecular mechanisms. First, we developed and implemented a mathematical model to capture the phenomenology of drug accumulation in our experiments, for example in terms of the measured lag in drug uptake and the time-varying uptake rate, without making assumptions regarding underlying biological mechanisms (e.g. positive feedback). Briefly, this model describes drug accumulation based on two coupled ordinary differential equations. The first equation describes drug accumulation in terms of uptake, which proceeds at a time-varying rate, and drug loss (due to efflux or degradation or dilution via growth [*Rybenkov et al., 2021*]), which we assume to be a first order reaction with rate constant $d_c$. The second equation describes how the drug uptake rate changes over time to take into account that the experimentally measured drug uptake is not always constant. Here we assume a state of uptake (parameter $k_1$, which switches on with a time delay; parameter $t_0$); a linear decay term (parameter $d_r$); as well as an adaptive inhibitory effect (parameter $k_2$) of the intracellular drug concentration on the uptake rate (allowing us to capture the dip we observe in some single-cell trajectories in *Figure 1—figure supplement 3*). We used this model to fit our single-cell *E. coli* data on the accumulation of all the above investigated drugs apart from vancomycin. This allowed us to compare and contrast the accumulation kinetic parameters above for the different antibiotics, since we used the same probe concentration for each drug (46 µg mL$^{-1}$) and all drugs were tested against the same clonal *E. coli* population. For vancomycin we found poor fitting for the majority of cells (195 out of 241 cells), as the fluorescent signal remained indistinguishable from the background, due to low cellular uptake (*Figure 1—figure supplement 2H*).

Membrane targeting antibiotic probes displayed on average faster accumulation onset ($t_0$=306, 364, and 571 s for tachyplesin, polymyxin B, and octapeptin, respectively) compared to antibiotics with an intracellular target ($t_0$=437, 2525, 3608, and 6,614 s for linezolid, trimethoprim, ciprofloxacin, and roxithromycin, respectively, *Figure 1—figure supplement 13* and *Figure 1—source data 1*). Remarkably, we found notable cell-to-cell differences in $t_0$ across all investigated drugs with a maximum CV of 209% for polymyxin B, and a minimum CV of 25% for roxithromycin (*Figure 1—figure supplement 13*), further confirming the presence of phenotypic variants with delayed antibiotic accumulation. It is also worth noting that linezolid displayed an accumulation onset value closer to the one recorded for antibiotics with a membrane target compared to the one measured for antibiotics with

an intracellular target. However, when considering the other five accumulation parameters linezolid displayed values in line with those measured for antibiotics with an intracellular target (see below).

Membrane targeting antibiotic probes also displayed, on average, steeper rates of uptake ($k_1$=260, 229, and 93 a.u. s$^{-2}$ for tachyplesin, polymyxin B, and octapeptin, respectively) compared to antibiotics with an intracellular target ($k_1$=4.4, 1.6, 0.9, and 0.3 a.u. s$^{-2}$ for roxithromycin, linezolid, ciprofloxacin, and trimethoprim, respectively, *Figure 1—figure supplement 13* and *Figure 1—source data 1*). Also, $k_1$ was heterogeneous across all drugs investigated with a maximum CV of 124% for roxithromycin and a minimum CV of 37% for trimethoprim (*Figure 1—figure supplement 13*), further confirming the presence of phenotypic variants with slow antibiotic uptake.

Membrane targeting antibiotic probes also displayed, on average, higher steady-state saturation levels ($F_{max}$ =2,597, 2,357, and 2,264 a.u. for tachyplesin, octapeptin, and polymyxin B, respectively) compared to antibiotics with an intracellular target ($F_{max}$ =1,034, 512, 253, and 180 a.u. for roxithromycin, linezolid, trimethoprim, and ciprofloxacin, respectively, *Figure 1—figure supplement 13* and *Figure 1—source data 1*). $F_{max}$ was also heterogeneous with a maximum CV of 55% for roxithromycin and a minimum CV of 9% for octapeptin (*Figure 1—figure supplement 13*) further confirming the presence of phenotypic variants with reduced antibiotic accumulation. For brevity, the second order kinetic parameters $k_2$, $d_r$, and $d_c$ are reported and discussed only in *Figure 1—figure supplement 14* and in *Figure 1—source data 1*.

The finding that accumulation of membrane targeting probes happens earlier, faster and to a greater extent than probes with an intracellular target can be easily rationalised considering that the latter probes need to cross the gram-negative double membrane. This represents a very good validation of our combined experimental and theoretical approach. However, the large heterogeneity in the kinetic parameters describing the accumulation of all probes, due to phenotypic variants with delayed or reduced accumulation, was instead unexpected. Additionally, the finding that roxithromycin simultaneously displayed the most delayed accumulation onset but also the steepest rate of uptake and highest steady-state saturation levels, across antibiotic probes with intracellular targets, was also unexpected. These data corroborate the hypothesis that multiple mechanisms must be involved in intracellular antibiotic accumulation at the level of the individual cell (*Rybenkov et al., 2021*), a point which we expand on below.

Next, we used the inferred accumulation kinetic parameters to test the hypothesis that phenotypic variants within a clonal population specialise to reduce antibiotic accumulation. When we pooled together the data for all the antibiotics tested against *E. coli*, we found a strong negative correlation between $t_0$ and $k_1$ and $t_0$ and $F_{max}$, but a strong positive correlation between $k_1$ and $F_{max}$ (*Figure 1— figure supplement 15A-C*, Pearson coefficients r=–0.40,–0.27, and 0.65, respectively, p<0.0001). The negative correlations between $t_0$ and $k_1$ and $t_0$ and $F_{max}$ across the bacterial population were not due to negative correlations for each individual cell. In fact, we found that for 86 and 79% of cells across all antibiotic treatments there was a positive correlation between $t_0$ and $k_1$ and between $t_0$ and $F_{max}$. In contrast, at the population level we found a significantly negative correlation between $t_0$ and $k_1$ for the accumulation of polymyxin B, octapeptin, and roxithromycin probes and a significantly negative correlation between $t_0$ and $F_{max}$ for the accumulation of polymyxin B, octapeptin, linezolid, and trimethoprim probes. Finally, we found a significantly positive correlation between $k_1$ and $F_{max}$ for the accumulation of polymyxin B, ciprofloxacin, and roxithromycin probes (*Appendix 1—table 2*). The latter correlation was partially imposed by the definition of $F_{max}$ in the model. In fact, we found that 78% of the cells displayed a positive correlation between these two parameters at the single-cell level. These strong correlations show that the bacteria, which start accumulating drugs later also display, slow uptake and low saturation levels. This statistical analysis also reveals that it is possible to rapidly identify phenotypic variants displaying reduced antibiotic accumulation by inferring the whole set of kinetic parameters from a smaller subset (e.g. by inferring $F_{max}$ from $t_0$ and $k_1$, the latter two can be measured significantly faster).

Furthermore, we also used our mathematical framework to test the hypothesis that treatment with each antibiotic gives rise to a unique accumulation profile. Using statistical classification with only two kinetic parameters ($t_0$ and $k_1$, i.e. the two parameters that can be rapidly measured experimentally), we found that treatment with membrane targeting probes is correctly classified against treatment with intracellular targeting probes with 99% accuracy (1075 cells analysed, *Figure 1—figure supplement 15D*). Moreover, treatment with polymyxin B, tachyplesin, or octapeptin was correctly classified

among treatments with the other two membrane targeting probes with 77, 76, and 64%, respectively (*Figure 1—figure supplement 15E-G*). Finally, treatment with linezolid, trimethoprim, ciprofloxacin, or roxithromycin was correctly classified among treatments with the other three intracellular targeting probes with 97, 84, 64, and 86% accuracy, respectively (*Figure 1—figure supplement 15H-K*). It is worth noting that we obtained similar levels of accuracy when we run such statistical classifications using the full set of kinetic accumulation parameters (i.e. $t_0$, $k_1$, $k_2$, $d_r$, and $d_c$), further demonstrating that measuring only $t_0$ and $k_1$ provides a good description of the antibiotic accumulation process.

Taken together, these data suggest the existence of a unique accumulation pattern for the specific antibiotic in use and could be employed in combination with existing single-cell microfluidic platforms to rapidly phenotype drug sensitivity (*Baltekin et al., 2017*; *Bakshi et al., 2021*; *Bergmiller et al., 2017*), ultimately in clinical antibiotic testing.

## Phenotypic variants with reduced antibiotic accumulation survive antibiotic treatment

Next, we hypothesised that phenotypic variants displaying reduced antibiotic accumulation also better survive antibiotic treatment, the correlation between antibiotic uptake and efficacy remaining poorly investigated (*Rybenkov et al., 2021*). We decided to focus on the macrolide roxithromycin since a large number of phenotypic variants displayed reduced roxithromycin accumulation (*Figure 1—figure supplement 3*). When we measured the elongation rate of individual cells while they were being dosed with roxithromycin-NBD dissolved in LB, we found two distinct cellular responses. While the majority of cells stopped growing during drug exposure, some phenotypic variants within the same clonal *E. coli* population continued elongating for the entire duration of drug treatment (see representative trajectories in *Figure 2A and B* and in *Figure 2—source data 1*).

Furthermore, there were significant cell-to-cell differences in the time at which cells stopped growing (*Figure 2A*). Notably, this time coincided with the onset in roxithromycin-NBD accumulation ($t_0$, indicated by circles and arrows in *Figure 2A*), suggesting a link between lag in drug uptake and lag in cell growth. However, other mechanisms might contribute to lag in drug uptake including a positive feedback loop in drug binding or positive feedback between efflux and drug accumulation (*Le et al., 2021*). Phenotypic variants that continued growing instead did not accumulate roxithromycin-NBD for the entire duration of the treatment (*Figure 2B*). When we formally analysed the entire dataset, we found a strong positive correlation between the onset of roxithromycin-NBD accumulation and the average elongation rate during exposure to roxithromycin-NBD ($r=0.49$, ***, *Figure 2C*, and *Figure 2—source data 2*). Moreover, bacteria that accumulated roxithromycin-NBD displayed a drastically reduced elongation rate after roxithromycin-NBD accumulation started compared to their elongation rate before uptake (**** paired *t*-test, *Figure 2D*, and *Figure 2—source data 3*). Phenotypic variants that did not accumulate roxithromycin-NBD instead displayed an elongation rate that was not significantly different compared to the elongation rate of bacteria that had not yet started taking up roxithromycin-NBD (ns unpaired *t*-test, *Figure 2D*). Finally, phenotypic variants that did not accumulate roxithromycin-NBD displayed an elongation rate that was significantly higher compared to the elongation rate of bacteria that had started taking up roxithromycin-NBD (**** unpaired *t*-test, *Figure 2D*). Moreover, we also found a significantly positive correlation between single-cell elongation rate before treatment and single-cell elongation rate during treatment ($r=0.34$, *, *Figure 2—figure supplement 1A*), although, as expected, the average elongation rate significantly decreased after roxithromycin-NBD addition ($5.2\pm2.9$ µm h$^{-1}$ vs $3.7\pm2.3$ µm h$^{-1}$, before and after drug addition, respectively, ****, *Figure 2—figure supplement 1A*, and *Figure 2—source data 4*). Finally, to further verify that these findings were not due to drug labelling, we performed these experiments with unlabelled roxithromycin. We found that the drug autofluorescence detected from the bacteria was not distinguishable from the background (i.e. the fluorescence detected from channels that did not contain bacteria). We confirmed a significantly positive correlation between single-cell elongation rate before treatment and single-cell elongation rate during treatment ($r=0.47$, ***, *Figure 2—figure supplement 1B* and *Figure 2—source data 5*, a point on which we expand below). We also found large single cell growth variability in the presence of labelled and unlabelled roxithromycin (CV of 59 and 44%, respectively).

Taken together these data demonstrate that cell-to-cell differences in drug accumulation are strongly linked with heterogeneity in antibiotic efficacy, prompting us to investigate the mechanisms underlying phenotypic variants with delayed or reduced antibiotic accumulation.

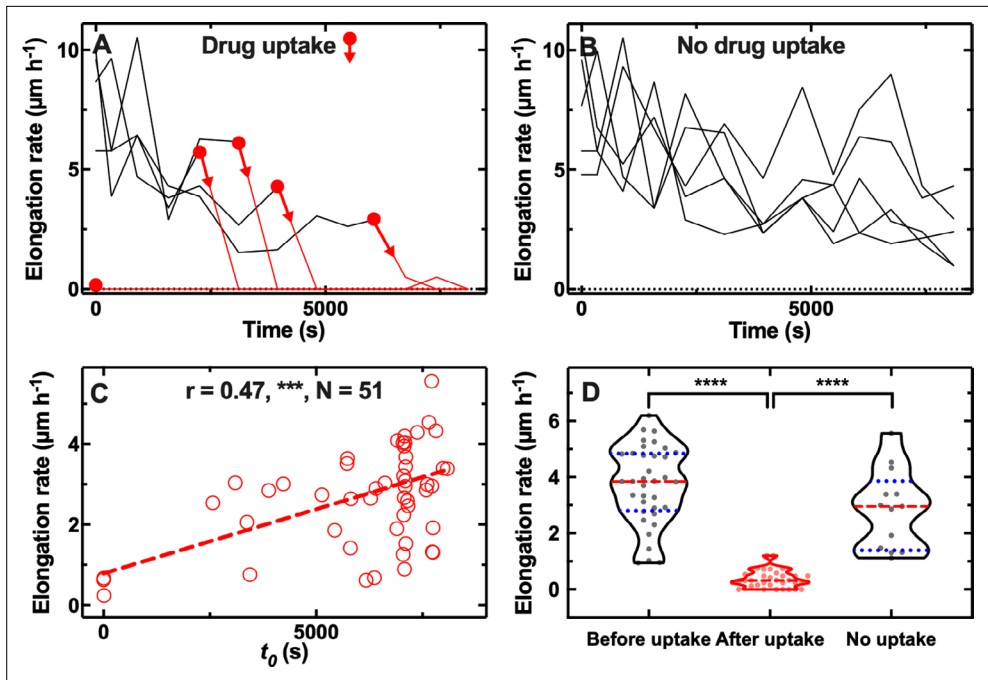

**Figure 2.** Correlation between antibiotic efficacy and antibiotic accumulation. Temporal patterns of elongation rate during exposure to the fluorescent derivative of roxithromycin for (**A**) five representative *E. coli* bacteria that accumulated the drug and (**B**) five representative *E. coli* bacteria that did not accumulate the drug. The fluorescent derivative of roxithromycin was delivered at t=0 at a concentration of 46 µg mL⁻¹ and was dissolved in lysogeny broth (LB), circles and arrows indicate $t_0$, the time point at which each bacterium started to accumulate the drug (i.e. bacterial fluorescence signal became distinguishable from the background). (**C**) Correlation between each bacterium $t_0$ and its average elongation rate throughout exposure to the fluorescent derivative of roxithromycin (i.e. 0<t< 8100 s). r is the Pearson coefficient quantifying the correlation above, ***: p-value<0.001, N=52 bacteria. (**D**) Average elongation rates for bacteria that had not yet started (before uptake) or had started (after uptake) accumulating the fluorescent derivative of roxithromycin, as well as for bacteria that did not accumulate the drug (no uptake). The red dashed and blue dotted lines within each violin plot represent the median and quartiles of each data set, respectively. Paired *t*-test between elongation rates before and after onset in accumulation: ****: p-value<0.0001, N=36 pairs. Unpaired *t*-test between the elongation rates of bacteria that did not take up the drug compared to the elongation rate of bacteria that had not yet started taking up the drug: not significant, p-value=0.07, N=13 and 36 bacteria, respectively. Unpaired *t*-test between the elongation rates of bacteria that did not take up the drug compared to the elongation rate of bacteria that had started taking up the drug: ****: p-value<0.0001, N=13 and 36 bacteria, respectively.

The online version of this article includes the following source data and figure supplement(s) for figure 2:

**Source data 1.** Single-cell elongation rates during roxithromycin treatment.

**Source data 2.** Correlation between drug accumulation and efficacy.

**Source data 3.** Average elongation rates for bacteria that had not yet started ccumulating the fluorescent derivative of roxithromycin.

**Source data 4.** Average elongation rates for bacteria that had started accumulating the fluorescent derivative of roxithromycin.

**Source data 5.** Average elongation rates for bacteria that did not accumulate the fluorescent derivative of roxithromycin.

**Figure supplement 1.** Interdependence between single-cell elongation rate before treatment and single-cell elongation rate during exposure to (**A**) roxithromycin-NBD and (**B**) unlabelled roxithromycin.

## The microcolony architecture affects heterogeneity in antibiotic accumulation

First, we tested the hypothesis that these phenotypic variants reduced antibiotic accumulation because of the presence of other bacteria (i.e. screening cells) between them and the main microfluidic

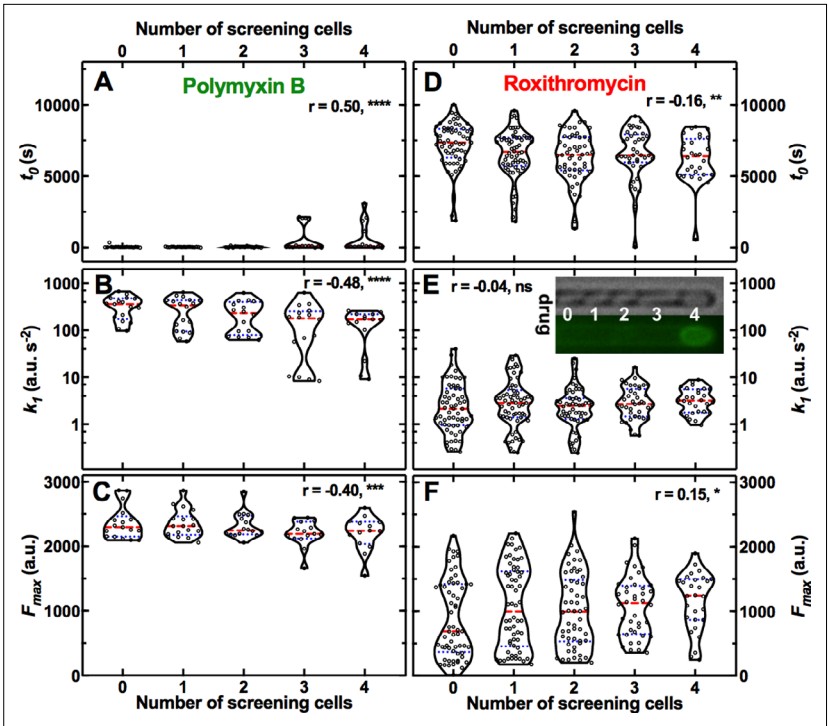

**Figure 3.** Effect of the presence of screening cells on the accumulation of antibiotics in single cells. Dependence of the kinetic parameters $t_0$, $k_1$, and $F_{max}$ for the accumulation of fluorescent derivatives of polymyxin B (**A–C**) and roxithromycin (**D–F**) on the number of screening cells between the bacterium under investigation and the main microfluidic chamber where the drug is continuously injected. Each data point is the value of a kinetic parameter inferred for an individual bacterium from the data in *Figure 1—figure supplement 2* using our mathematical model, N=103 and 265 for polymyxin B and roxithromycin, respectively. The red dashed and blue dotted lines within each violin plot represent the median and quartiles of each data set, respectively. r is the Pearson coefficient quantifying the correlation between each inferred kinetic parameter and the number of screening cells in front of each bacterium. ns: not significant correlation, *: p-value<0.05, **: p-value<0.01, ***: p-value<0.001, ****: p-value<0.0001. Inset: representative brightfield and fluorescence images illustrating, from left to right, a bacterium screened by 0, 1, 2, 3, and 4 cells, respectively; roxithromycin-NBD was injected in the main microfluidic chamber in the left-hand side of the image and diffused from left to right. The fluorescence image shows early roxithromycin-NBD accumulation in the bacterium screened by the highest number of cells.

The online version of this article includes the following source data and figure supplement(s) for figure 3:

**Source data 1.** Impact of microcolony architecture on drug accumulation.

**Source data 2.** Simulations of antibiotic diffusion and absorption.

**Figure supplement 1.** Simulations of antibiotic diffusion and absorption.

chamber, where the drug is injected. To test this hypothesis, we classified our data in subpopulations of bacteria that had 0, 1, 2, 3, or 4 screening cells between themselves and the main microfluidic chamber (see Inset in *Figure 3E* where the drug diffuses from left to right).

For polymyxin B we observed that increasing the number of screening cells increased $t_0$ while reducing $k_1$ and $F_{max}$ (Pearson correlation coefficient r=0.50,–0.48, and –0.40,****, **** and ***, respectively, *Figure 3A–C* and *Figure 3—source data 1*). Moreover, octapeptin and tachyplesin displayed strong negative correlation between $k_1$ and the number of screening cells (r=–0.63 and –0.67, respectively, ****); octapeptin also displayed a strong positive correlation between $t_0$ and the number of screens (r=0.71, ****). These data were in accordance with our hypothesis that screening cells transiently decrease the pool of drug molecules available for screened cells until the bacteria closer to the main chamber reach antibiotic accumulation saturation levels. These data could explain the large heterogeneity in $t_0$ measured for such membrane-targeting probes (*Figure 1—figure supplement 13*). In contrast with our hypothesis, for roxithromycin we found that increasing the number of screens in front of a cell reduced $t_0$ and increased $F_{max}$ (r=–0.16 and 0.15, **, and *, respectively, *Figure 3D–F*

and *Figure 3—source data 1*). Moreover, both ciprofloxacin and linezolid displayed a strong negative correlation between $t_0$ and the number of screens ($r$=–0.53 and –0.28, **** and ***, respectively); ciprofloxacin also displayed a strong positive correlation between $k_1$ and the number of screens ($r$=0.32, ***).

Delayed accumulation of membrane targeting drugs in bacteria screened by other cells could be explained by a transient reduction in the extracellular drug concentration around these bacteria (compared to the concentration in the main microfluidic chamber) due to rapid drug binding to the membranes of screening cells. In accordance with this hypothesis, when we run 2D numerical simulations of drug diffusion in channels hosting bacteria with a high drug absorption rate ($\gamma$=0.2 mol m$^{-2}$ s$^{-1}$, see Methods), we found a gradient in extracellular drug concentration along the channel length: for the first 90 min post drug addition, the concentration was highest around the bacterium without screens and lowest around the bacterium with four screens (*Figure 3—figure supplement 1A*). On the contrary, in the presence of bacteria with a low absorption rate ($\gamma$=0.002 mol m$^{-2}$ s$^{-1}$), the extracellular drug concentration equilibrated along the channel length within 2 min post drug addition to the device (*Figure 3—figure supplement 1C*). Accordingly, in the presence of bacteria with high absorption rate, the intracellular drug concentration (that we simply modelled as concentration at the bacterial surface) reached saturation levels in the bacterium without screens within minutes post drug addition, whereas the bacterium with 4 screens reached saturation levels 90 min post drug addition (*Figure 3—figure supplement 1B* and *Figure 3—source data 2*). Conversely, bacteria with low absorption rate accumulated the drug independently on the number of screens (*Figure 3—figure supplement 1D*). Therefore, according to these simplified 2D transport simulations (i.e. we do not take into account neither efflux nor transport across the gram-negative double barrier), delayed accumulation of membrane targeting drugs in bacteria screened by other cells is due to a transient reduction in the extracellular drug concentration around these bacteria, whereas other mechanisms must underpin increased roxithromycin accumulation in screened bacteria and this phenomenon should be investigated further in future studies. It is worth noting that these findings were not dictated by oxygen limitation or low metabolic activity as in the case of biofilms (*Walters III et al., 2003*). In fact, we (*Łapińska et al., 2019*; *Glover et al., 2022*) and others (*Wang et al., 2010*) have previously demonstrated that nutrients, including oxygen and metabolites, uniformly distribute across the whole length of bacteria hosting channels in our microfluidic device.

It is also worth noting that mechanisms other than the microcolony architecture must underlie phenotypic variants with reduced antibiotic accumulation. In fact, we registered significant cell-to-cell differences in antibiotic accumulation even within the same subpopulation of bacteria with the same number of screening cells; these differences were more pronounced for antibiotic with intracellular targets compared to membrane targeting antibiotics (e.g. roxithromycin and polymyxin B, respectively, in *Figure 3*).

## Cell-to-cell differences in growth rate before treatment contribute to heterogeneity in antibiotic accumulation

In order to further dissect the mechanisms underlying phenotypic variants with reduced antibiotic accumulation, we took advantage of continuous live-cell imaging to track individual bacteria for the 2 hr growth period in LB before incubation in each antibiotic. This permitted us to investigate links between each bacterium's growth and its capability to avoid or delay antibiotic accumulation. We investigated the correlation between elongation rates before treatment (averaged over all the values obtained during the 2 hr growth period) and the kinetic parameters describing the accumulation of two representative membrane-targeting antibiotics, i.e., octapeptin and tachyplesin, and two representative antibiotics with intracellular targets, i.e., trimethoprim and roxithromycin.

We did not find any significant correlation between single-cell elongation rate before treatment and any of the kinetic parameters describing the accumulation of octapeptin and trimethoprim (*Figure 4—figure supplement 1A-C and 1G-I*, respectively, and *Figure 4—source data 1*; *Figure 4—source data 3*). However, we found a positive correlation between single-cell elongation rate before treatment and $k_1$ for tachyplesin ($r$=0.59, **, *Figure 4—figure supplement 1E*, and *Figure 4—source data 2*), suggesting that the latter accumulated faster in fast growing cells. On the contrary, for roxithromycin, we found a significantly positive correlation between single-cell elongation rate before treatment and $t_0$ and a significantly negative correlation between single-cell elongation rate

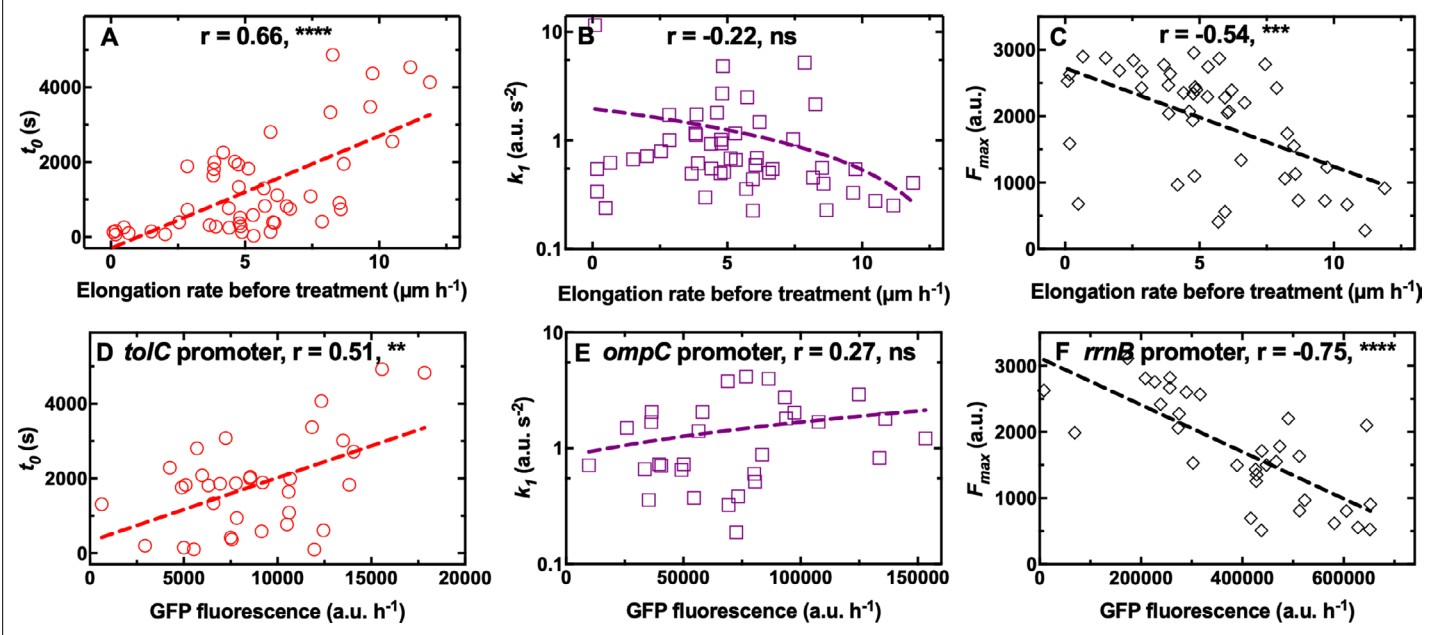

**Figure 4.** Differential cell growth and expression of key molecular pathways prior antibiotic treatment contributes to heterogeneity in roxithromycin accumulation. (**A–C**) Correlation between the single-cell kinetic parameters $t_0$, $k_1$ and $F_{max}$ describing the accumulation of roxithromycin-NBD and the bacterial elongation rate during the 2 hr growth period preceding antibiotic treatment (see Methods). Measurements were carried out on N=50 individual *E. coli*, collated from biological triplicate, before and after exposure to 192 μg mL$^{-1}$ roxithromycin-NBD dissolved in M9. (**D–F**) Correlation between single-cell green fluorescent protein (GFP) fluorescence, as a proxy for the expression of *tolC*, *ompC*, and *rrnB* promoters, and single-cell kinetic parameters $t_0$, $k_1$ and $F_{max}$ describing the accumulation of roxithromycin-DMACA (at an extracellular concentration of 192 μg mL$^{-1}$). r is the Pearson coefficient quantifying the correlation between each inferred kinetic parameter and the corresponding elongation rate of each cell. ns: not significant correlation, **: p-value<0.01, ***: p-value<0.001, ****: p-value<0.0001. Dashed lines are linear regressions to the data. Measurements were carried out on N=34, 30, and 33 individual *E. coli* collated from biological triplicate for the *tolC*, *ompC*, and *rrnB* reporter strains, respectively.

The online version of this article includes the following source data and figure supplement(s) for figure 4:

**Source data 1.** Correlation between octapeptin accumulation and cell growth state.

**Source data 2.** Correlation between tachyplesin accumulation and cell growth state.

**Source data 3.** Correlation between trimethoprim accumulation and cell growth state.

**Source data 4.** Correlation between roxithromycin accumulation and cell growth state.

**Source data 5.** Correlation between the expression of *tolC* and roxithromycin accumulation.

**Source data 6.** Correlation between the expression of *ompC* and roxithromycin accumulation.

**Source data 7.** Correlation between the expression of *rrnB* and roxithromycin accumulation.

**Source data 8.** Correlation between the expression of *rrnH* and roxithromycin accumulation.

**Figure supplement 1.** Correlation between drug accumulation and cell growth state.

**Figure supplement 2.** Differential cell growth contributes to heterogeneity in roxithromycin accumulation.

**Figure supplement 3.** Distribution of single-cell elongation rates during roxithromycin treatment.

**Figure supplement 4.** Negative correlation between ribosomal expression and roxithromycin accumulation.

before treatment and $F_{max}$ (r=0.66 and –0.54, **** and ***, respectively, *Figure 4A and C*, and *Figure 4—source data 4*), but no correlation with cell size (*Figure 1—figure supplement 6*). These data suggest that roxithromycin accumulated more slowly and to a lesser extent in bacteria that were growing faster prior to antibiotic treatment. The measured variability in the elongation rate before antibiotic treatment was larger compared to that measured in previous reports (*Baltekin et al., 2017*; *Wang et al., 2010*). This discrepancy is due to the fact that we started to measure single-cell elongation rates immediately upon loading an aliquot of stationary phase *E. coli* in the mother machine. In contrast previous studies either did not measure elongation rates of the first 10 generations upon loading *E. coli* in the mother machine (*Wang et al., 2010*) or precultured *E. coli* for 2 hr before loading in the mother machine (*Baltekin et al., 2017*). Therefore, our measurements capture heterogeneous

growth resumption from stationary phase (*Levin-Reisman et al., 2010*; *Jõers and Tenson, 2016*), with a minority of bacteria that did not divide in the 2 hr growth prior to antibiotic exposure (i.e. 10%, non dividing subgroup *Figure 4—figure supplement 2A-C*), a few bacteria that divided once (i.e. 30%, slow dividing subgroup, *Figure 4—figure supplement 2D-F*), and the majority of bacteria that divided two times (i.e. 60%, fast dividing subgroup, *Figure 4—figure supplement 2G-I*). Elongation rates within bacteria that divided once fell within 1.5 and 5 μm h$^{-1}$, whereas elongation rates within bacteria that divided two times fell within 4 and 12 μm h$^{-1}$, with a within subgroup variation of 3–4 folds in line with previous reports (*Baltekin et al., 2017*; *Wang et al., 2010*). Importantly, cells within both the slow and fast dividing subgroups displayed a significant correlation between elongation rates before treatment and $t_0$ or $F_{max}$ (*Figure 4—figure supplement 2*).

Taken together these data suggest that phenotypic variants displaying delayed or reduced roxithromycin accumulation are bacteria that grow faster before antibiotic treatment starts. These novel findings are surprising considering that phenotypic survival to antibiotics has traditionally been linked to slow growth, low metabolic activity, and bacterial dormancy prior to antibiotic treatment (*Balaban et al., 2004*; *Balaban et al., 2013*; *Lewis, 2007*). In contrast, here we show that fast growth prior to antibiotic treatment facilitates delayed roxithromycin accumulation as well as reducing the amount of macrolide accumulating in individual bacteria at steady state. It is worth noting that growth in cell volume during roxithromycin treatment might help bacteria in diluting the intracellular drug concentration. In fact, we measured a 3-fold variation in elongation rate (and thus in cell volume considering that the other two cell dimensions are physically constrained in all cells due to the device geometry), with the slowest and fastest bacterium displaying an elongation rate of 0.7 and 2.3 μm h$^{-1}$, respectively, during treatment with roxithromycin at a growth inhibitory concentration (i.e. 192 μg mL$^{-1}$, *Figure 4—figure supplement 3*). Therefore, the measured variation in roxithromycin accumulation is due in part to dilution of the intracellular drug concentration via differential cell growth. This effect is accounted for in our phenomenological mathematical model via the rate constant $d_c$, which describes drug loss through efflux, degradation or dilution via growth. Moreover, other factors must also play a role in the measured cell-to-cell differences in roxithromycin accumulation. In fact, these differences include a 160-fold variation in $t_0$ (30 s $< t_0 <$ 4900 s), a 60-fold variation in $k_1$ (0.2 a.u. s$^{-2}$ $< k_1 <$ 12 a.u. s$^{-2}$) and a 12-fold variation in $F_{max}$ (250 a.u. $< F_{max} <$ 3000 a.u.) far larger than the measured variation in elongation rate.

## Single-cell ribosome and efflux pump abundance contributes to heterogeneity in macrolide accumulation

In order to determine other mechanisms underpinning phenotypic variants with reduced roxithromycin accumulation, we investigated some of the key molecular pathways underlying antibiotic accumulation (*Rybenkov et al., 2021*). We hypothesised that heterogeneity in $t_0$ could be linked to cell-to-cell differences in the capability to pump antibiotics out from the cell, thus delaying the onset of accumulation. *tolC*, which encodes the outer membrane channel of the multidrug efflux pump AcrAB-TolC and the macrolide efflux pump MacAB-TolC (*Rybenkov et al., 2021*), was the most highly expressed efflux pump related gene according to our transcriptomic data of *E. coli* cultures growing on LB for a period of 2 hr after dilution of an overnight culture (*Appendix 1—table 3*; *Smith et al., 2018*). Therefore, we used a *tolC* transcriptional reporter strain (*Bamford et al., 2017*) to establish a link between $t_0$ and *tolC* expression during the 2 hr growth period before exposure to roxithromycin. In line with our hypothesis above, we found a positive correlation between the expression of *tolC* and $t_0$ (r=0.51, **, *Figure 4D*, and *Figure 4—source data 5*).

Next, we hypothesised that heterogeneity in the rate of drug uptake $k_1$ could be ascribed to cell-to-cell differences in the expression of outer membrane porins allowing antibiotic passage across the outer membrane. *ompC*, which encodes the outer membrane protein OmpC facilitating influx of several antibiotics (*Rybenkov et al., 2021*; *Nikaido, 2003*), was the most highly expressed outer membrane protein encoding gene according to our transcriptomic data at the population level (*Appendix 1—table 3*, *Smith et al., 2018*). In contrast with our hypothesis, we did not find a significant correlation between *ompC* expression and $k_1$ (r=0.27, ns, *Figure 4E* and *Figure 4—source data 6*).

Finally, we hypothesised that saturation levels in roxithromycin accumulation could depend on the ribosomal content (i.e. the drug target) at the single-cell level. Accordingly, we found a strong

negative correlation between the expression of the ribosomal promoters *rrnB* and *rrnH* [two ribosomal promoters with different level of expression (*Maeda et al., 2015*) and routinely used as reporters for ribosomal activity (*Panlilio et al., 2021*) and $F_{max}$ ($r$=–0.75 and –0.68, ****, *Figure 4F*, *Figure 4—source data 7*, *Figure 4—figure supplement 4*, and *Figure 4—source data 8*, respectively).

Taken together these data shed light on the molecular mechanisms underpinning the observed heterogeneity in the intracellular accumulation of the macrolide roxithromycin: fast growing variants reduce the intracellular accumulation of roxithromycin, and thus better survive treatment with this drug, via elevated ribosomal content and, to a lesser extent, higher expression of efflux pumps. These data suggest that reduced metabolism and dormancy might not always represent the best bacterial strategy for overcoming antibiotic challenge (*Balaban et al., 2004*; *Balaban et al., 2013*; *Lewis, 2007*; *Otto, 2021*).

## External manipulation of the heterogeneity in antibiotic accumulation

Building on the molecular understanding gained above, we then set out to establish whether phenotypic variants displaying reduced roxithromycin accumulation could be suppressed via genetic or chemical manipulation. In order to do so, we employed a *ΔtolC* knockout mutant and found that, when investigating roxithromycin accumulation, $t_0$ was significantly lower and $k_1$ was significantly higher in the *ΔtolC* mutant compared to the parental strain (*Figure 5A and B*, respectively, and *Figure 5—source data 1*).

However, we also found *ΔtolC* phenotypic variants with reduced roxithromycin accumulation and even higher levels of heterogeneity in the three kinetic parameters for the *ΔtolC* mutant compared to the parental strain (CV of 27 vs 25%, 114 vs 80%, 72 vs 62% for $t_0$, $k_1$, and $F_{max}$, respectively, *Figure 5*). Furthermore, the *ΔtolC* knockout strain did not display a significantly longer average doubling time during roxithromycin treatment compared to the parental strain (*Figure 1—figure supplement 5*). Therefore, the measured impact of the absence of *tolC* on roxithromycin accumulation (i.e. significantly shorter $t_0$ and larger $k_1$) cannot be ascribed to dilution via cell doubling.

Next, we employed a *ΔompC* knockout mutant and found that roxithromycin accumulation was not significantly different compared to

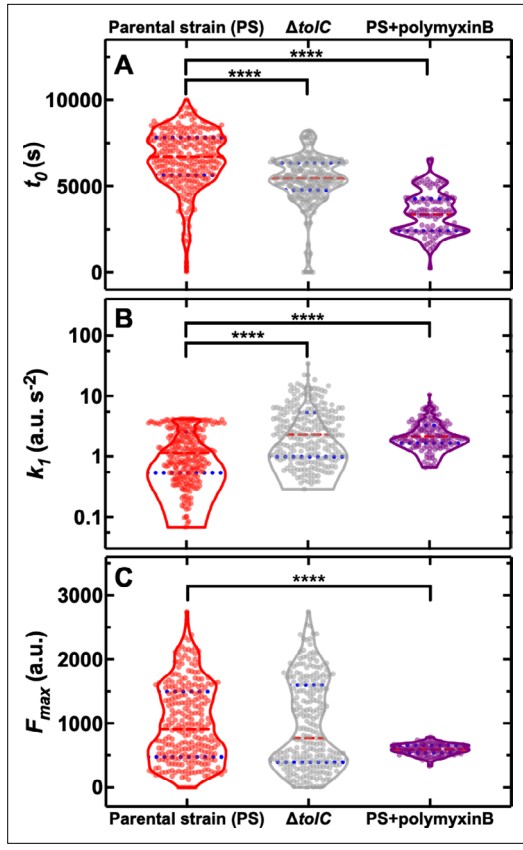

**Figure 5.** Genetic and chemical manipulation of heterogeneity in drug accumulation. Distributions of single-cell values for the kinetic parameters (**A**) $t_0$, (**B**) $k_1$ and (**C**) $F_{max}$ describing the accumulation of the fluorescent derivative of roxithromycin (at 46 µg mL$^{-1}$ in M9) in the *E. coli* BW25113 parental strain (PS), the knockout mutant *ΔtolC* and the parental strain co-treated with unlabelled polymyxin B at 1 µg mL$^{-1}$ extracellular concentration. The red dashed and blue dotted lines within each violin plot represent the median and quartiles of each data set, respectively. ****: p-value<0.0001. N=262, 241, and 116 individual parental strain *E. coli* treated with the roxithromycin probe, *ΔtolC E. coli* treated with the roxithromycin probe and parental strain *E. coli* co-treated with the roxithromycin probe and 1 µg mL$^{-1}$ unlabelled polymyxin B.

The online version of this article includes the following source data and figure supplement(s) for figure 5:

**Source data 1.** Genetic and chemical manipulation of heterogeneity in drug accumulation.

**Source data 2.** Genetic manipulation of heterogeneity in drug accumulation.

**Figure supplement 1.** Distributions of single-cell values for the kinetic parameters (**A**) $t_0$, (**B**) $k_1$ and (**C**) $F_{max}$ describing the accumulation of the fluorescent derivative of roxithromycin (at 46 µg mL$^{-1}$ in M9) in the *E. coli* BW25113 parental strain and the knockout mutant *ΔompC*. The red dashed and blue dotted

*Figure 5 continued on next page*

*Figure 5 continued*

lines within each violin plot represent the median and quartiles of each data set, respectively. None of the kinetic parameters was significantly different in the knockout mutant $\Delta ompC$ compared to the parental strain, p-values of 0.39, 0.69, and 0.41, respectively. N=262 and 100 individual bacteria for the parental and $\Delta ompC$ strain, respectively.

that measured for the parental strain (*Figure 5—figure supplement 1* and *Figure 5—source data 2*), thus corroborating the data above obtained using an *ompC* fluorescent reporter strain (*Figure 4E*). Therefore, we hypothesised that the composition and permeability of the lipid bilayer making up the bacterial outer membrane could underlie heterogeneity in roxithromycin accumulation. If this were true, the heterogeneity in roxithromycin accumulation could be chemically manipulated by using agents that permeabilise the outer membrane, such as polymyxin B (*Vaara, 1992*). Accordingly, when we treated the parental strain with roxithromycin-NBD at 46 μg mL$^{-1}$ in combination with unlabelled polymyxin B at 1 μg mL$^{-1}$ extracellular concentration, we found a significant decrease in the heterogeneity of $k_1$ and $F_{max}$ compared to roxithromycin-NBD treatment alone (CV of 59 vs 80%, 14 vs 62%, respectively, *Figure 5B and C*). Additionally, the accumulation dynamics of roxithromycin-NBD in the presence of unlabelled polymyxin B was significantly earlier and faster compared to that measured in the absence of polymyxin B (*Figure 5*). Taken together, these data suggest that phenotypic variants displaying reduced roxithromycin accumulation might have a significantly more impermeable outer membrane than phenotypically susceptible bacteria, possibly due to differences in lipid composition and packing and that targeting the outer membrane might be a viable avenue for suppressing variants with reduced intracellular antibiotic accumulation.

## Discussion

Bacterial slow growth has often been associated with decreased antibiotic susceptibility (*Balaban et al., 2004*; *Balaban et al., 2019*; *Pontes and Groisman, 2020*) with few exceptions (*Orman and Brynildsen, 2013*; *Peyrusson et al., 2020*). Moreover, a recent paper suggested that phenotypic variants accumulate lower levels of phenoxymethylpenicillin while being in a dormant state before treatment (*Pu et al., 2016*). In contrast, our data suggest that fast growth and elevated ribosomal content better prepare phenotypic variants for avoiding the intracellular accumulation of roxithromycin, a finding that could inform the design of antibiotic therapy using macrolides.

A linear correlation between ribosomal abundance and growth rate has previously been found via ensemble measurements obtained on exponentially growing *E. coli* supplied with nutrients of increasing quality in the absence of antibiotics (*Scott et al., 2010*). Our findings enrich the current understanding of the interdependence of cell growth and ribosomal content demonstrating that this correlation holds within an isogenic population homogeneously exposed to the same medium.

Previous ensemble measurements have demonstrated that fast growth on high quality nutrients decreases *E. coli* growth inhibition by antibiotics that irreversibly bind to ribosomes (such as roxithromycin [*Dinos et al., 2003*]) compared to slower growth on poor quality nutrients (*Greulich et al., 2015*). Here, we offer a mechanistic understanding of this unexpected finding, showing that reduced growth inhibition in fast growing cells is dictated by growth-dependent transport rates, as fast growing variants displayed reduced macrolide accumulation. Importantly, we demonstrated that this phenotypic response is found not only at the population-level (*Greulich et al., 2015*), but also within an isogenic population.

These new data can be rationalised by considering that in fast growing variants a fraction of leading actively translating ribosomes (*Dai et al., 2016*) escapes roxithromycin binding, while other ribosomes stall after accumulating roxithromycin. Drug-free active ribosomes continue to facilitate essential cellular processes including efflux that can reduce macrolide accumulation. Accordingly, we found that variants delaying roxithromycin accumulation also displayed a significantly higher expression of the efflux promoter *tolC* compared to bacteria that readily accumulated roxithromycin. Moreover, the deletion knockout $\Delta tolC$ displayed significantly earlier and faster accumulation of roxithromycin compared to the parental strain, confirming that roxithromycin is a substrate of the AcrAB- and MacAB-TolC efflux pumps (*Silver, 2016*). However, this mutant exhibited accumulation heterogeneity levels comparable to the parental strain. These data suggest that phenotypic variants reduce

antibiotic accumulation using processes other than efflux alone, in contrast with previous findings (*Pu et al., 2016*), and in accordance with our data on the key role played by heterogeneity in ribosomal abundance.

Our data also revealed a strong correlation between the accumulation of roxithromycin and the effect of this antibiotic on cell growth down to the scale of the individual cell. This suggests that phenotypic variants with reduced antibiotic accumulation could be an important factor contributing to phenotypic resistance to antibiotics (*Ackermann, 2015*; *Bamford et al., 2017*; *Balaban et al., 2019*; *Wilmaerts et al., 2019*). This new knowledge deepens our understanding of phenotypic resistance to antibiotics that is currently centred around target deactivation or modification (*Balaban et al., 2019*; *Gollan et al., 2019*; *Defraine et al., 2018*) with very little known about the correlation between antibiotic accumulation and antibiotic efficacy (*Rybenkov et al., 2021*; *Pu et al., 2016*).

Experimental evidence suggests that both macrolides and polymyxins use the self-promoted uptake pathway. Moreover, polymyxins have a higher affinity to the lipopolysaccharides (LPS) compared to macrolides and increase the permeability of the outer membrane to other freely diffusing antibiotic molecules (*Farmer et al., 1992*). Accordingly, we observed that OmpC, which is a major route of antibiotic influx via the hydrophilic pathway (*Delcour, 2009*), did not play a role in roxithromycin accumulation. Our data show instead that the phenotypic variants that avoid roxithromycin accumulation can be suppressed by delivering roxithromycin in combination with polymyxin B. Moreover, roxithromycin accumulated at lower saturation levels in the presence of polymyxin B as expected due to competitive binding to the LPS.

These data suggest that heterogeneity in roxithromycin accumulation could also be due to cell-to-cell differences in LPS composition. It is conceivable that phenotypic variants within the clonal population might have a decreased ethanolamine content. This would result in an increased negative charge of the LPS core and a decreased permeability to roxithromycin but not to polymyxin B (*Clark, 1984*) in accordance with our data. It is also conceivable that phenotypic variants within a clonal population might display esterification of the core-lipid A phosphates (*Peterson et al., 1987*). However, this would result in decreased permeability to both roxithromycin and polymyxin B in contrast with our data showing (i) comparatively smaller cell-to-cell differences in polymyxin B accumulation (beyond the heterogeneity generated by the microcolony architecture) and (ii) that adding polymyxin B suppresses the heterogeneity in roxithromycin accumulation. Finally, it has been suggested that macrolides use the hydrophobic pathway (*Vaara, 1993*). It is conceivable that phenotypic variants within the clonal population might display a higher expression of *lpxA* and thus reduced permeability to roxithromycin; however, this hypothesis remains to be tested.

We further demonstrate that the presence of phenotypic variants that avoid antibiotic accumulation is not dictated by the microcolony architecture (as represented by bacterial cell position within a microfluidic channel). However, our data are in agreement with previous work in clinical settings suggesting that macrolides, quinolones, and oxazolidinones are more effective within infecting biofilms compared to glycopeptides and polymyxins (*Walters III et al., 2003*; *Wu et al., 2015*). In fact, we demonstrate that antibiotics with intracellular targets accumulate more readily and to higher saturation levels in bacteria within the inner core of the colony. In contrast, membrane targeting drugs accumulate more readily, faster, and at higher saturation levels in bacteria at the outer rim of the colony. This drug-specific effect of colony architecture on drug accumulation must rely on growth-independent mechanism and efflux-independent mechanism. In fact, we did not find significant correlations between the position of a cell within the colony and neither the expression of *tolC*, *ompC*, or *rrnB* nor the bacterial elongation rate (p-value=0.13, 0.13, 0.46, and 0.34, respectively).

In conclusion, this work reveals hitherto unrecognised phenotypic variants that avoid antibiotic accumulation within bacterial populations. In contrast with the current consensus, we demonstrate that fast growing phenotypic variants avoid macrolide accumulation and survive treatment due to elevated ribosomal content. We further show that it is possible to eradicate phenotypic variants currently avoiding macrolide accumulation by using a roxithromycin-polymyxin combination therapy. These data give strength to recent evidence that administered doses of polymyxins can be lowered in combination therapies (*Brochado et al., 2018*) and demonstrate that roxithromycin could be repurposed against gram-negative bacteria. Finally, our novel single-cell approach reveals that each antibiotic is characterised by a unique accumulation pattern and thus could in future be employed to simultaneously characterise the accumulation and efficacy of new leading antibiotic compounds

(*Kepiro et al., 2020*; *Hammond et al., 2021*; *Nonejuie et al., 2013*; *Stokes et al., 2020*; *Cama et al., 2022*).

## Materials and methods

### Chemicals and cell culture

All chemicals were purchased from Fisher Scientific or Sigma-Aldrich unless otherwise stated. LB medium (10 g L$^{-1}$ tryptone, 5 g L$^{-1}$ yeast extract, and 0.5 g L$^{-1}$ NaCl) and LB agar plates (LB with 15 g L$^{-1}$ agar) were used for planktonic growth and setting up overnight cultures. Glucose-free M9-minimal media, used to dissolve fluorescent antibiotic derivatives was prepared using 5×M9 minimal salts (Merck), diluted as appropriate, with additional 2 mM MgSO$_4$, 0.1 mM CaCl$_2$, 3 µM thiamine HCl in Milli-Q water. Stock solutions of polymyxin B, octapeptin, tachyplesin, vancomycin, linezolid, roxithromycin, and trimethoprim were obtained by dissolving these compounds in dimethyl sulfoxide; ciprofloxacin instead was dissolved in 0.1 M HCl in Milli-Q water. These stock solutions were prepared at a concentration of 640 µg mL$^{-1}$. *E. coli* BW25113 was purchased from Dharmacon (GE Healthcare). *ompC*, *tolC*, *rrnH* and *rrnB* reporter strains of an *E. coli* K12 MG1655 promoter library (*Zaslaver et al., 2006*) were purchased from Dharmacon . Plasmids were extracted and transformed into chemically competent *E. coli* BW25113 as previously reported (*Henry and Brynildsen, 2016*). Deletion mutants *ΔtolC* and *ΔompC* were also purchased from Dharmacon. *S. aureus* ATCC 25923, *P. aeruginosa* PA14 flgK::Tn5(Tcr) (the deletion of the flagellum FlgK facilitated holding cells in the hosting channel thanks to the reduced bacterial motility) and *B. cenocepacia* K56-2 were kindly provided by A. Brown and S. van Houte. All strains were stored in 50% glycerol stock at –80 °C. Streak plates for each strain were produced by thawing a small aliquot of the corresponding glycerol stock every 2 weeks and plated onto LB agar. Overnight cultures were prepared by picking a single bacterial colony from a streak plate and growing it in 100 mL fresh LB medium on a shaking platform at 200 rpm and 37 °C for 17 hr.

### Synthesis of fluorescent derivatives of antibiotics

Fluorescent antibiotic derivatives from trimethoprim (*Phetsang et al., 2016*) (antifolate), linezolid (*Phetsang et al., 2014*) (oxazolidinone), ciprofloxacin (*Stone et al., 2019*) (fluoroquinolone), and roxithromycin (*Stone et al., 2020*) (macrolide) were prepared as previously described. Vancomycin (*Blaskovich et al., 2018*) (glycopeptide), polymyxin (*Gallardo-Godoy et al., 2016*), and octapeptin (*Velkov et al., 2018*) (both lipopeptides) and tachyplesin (*Edwards et al., 2017*) (antimicrobial peptide) analogues were designed and synthesised based on structure-activity-relationship studies and synthetic protocols reported in prior publications, introducing an azidolysine residue for the subsequent 'click' reactions with nitrobenzoxadiazole (NBD)-alkyne. Additionally, a fluorescent derivative of roxithromycin using the fluorophore dimethylamino-coumarin-4-acetate (DMACA) was synthesised and used only to determine the impact of labelling on single-cell antibiotic accumulation.

### Determination of minimum inhibitory concentration

Single colonies of *E. coli* BW25113 were picked and cultured overnight in cation-adjusted Mueller Hinton broth (CAMHB) at 37 °C, then diluted 40-fold and grown to OD$_{600}$=0.5. 60 µL of each antibiotic or fluorescent antibiotic derivative stocks were added to the first column of a 96-well plate. 40 µL CAMHB was added to the first column, and 30 µL to all other wells. 70 µL solution was then withdrawn from the first column and serially transferred to the next column until 70 µL solution withdrawn from the last column was discharged. The mid-log phase cultures (i.e. OD$_{600}$=0.5) were diluted to 10$^6$ colony forming units (c.f.u.) ml$^{-1}$ and 30 µL was added to each well, to give a final concentration of 5×10$^5$ c.f.u. ml$^{-1}$. Each plate contained 2 rows of 12 positive control experiments (i.e. bacteria growing in CAMHB without antibiotics) and two rows of 12 negative control experiments (i.e. CAMHB only). Plates were covered with aluminium foil and incubated at 37 °C overnight. The minimum inhibitory concentrations (MICs) of fluorescent derivatives of polymyxin B, octapeptin, tachyplesin, vancomycin, linezolid, roxithromycin, ciprofloxacin, trimethoprim, and each corresponding parental antibiotic against *E. coli* BW25113 were determined visually, with the MIC being the lowest concentration well with no visible growth (compared to the positive control experiments).

### Fabrication of the microfluidic devices

The mould for the mother machine microfluidic device was fabricated by Kelvin Nanotechnology using previously established multilevel photolithography processes (*Pagliara et al., 2007*). This mould

is equipped with six identical microfluidic networks that can be controlled simultaneously and independently to maximise experimental throughput. Each of these networks is equipped with approximately 6000 lateral microfluidic channels with width and height of 1 μm each and a length of 20 μm. These lateral channels are connected to a main microfluidic chamber that is 25 μm and 100 μm in height and width, respectively. Polydimethylsiloxane (PDMS) replicas of this device were realised as previously described (*Locatelli et al., 2016*). Briefly, a 10:1 (base:curing agent) PDMS mixture was cast on the mould and cured at 70 °C for 120 min in an oven. The cured PDMS was peeled from the epoxy mould and fluidic accesses were created by using a 0.75 mm biopsy punch (Harris Uni-Core, WPI). The PDMS chip was irreversibly sealed on a glass coverslip by exposing both surfaces to oxygen plasma treatment (10 s exposure to 30 W plasma power, Plasma etcher, Diener, Royal Oak, MI, USA). This treatment temporarily rendered the PDMS and glass hydrophilic, so immediately after bonding the chip was filled with 2 μL of a 50 mg/mL bovine serum albumin solution and incubated at 37 °C for 30 min, thus passivating the internal surfaces of the device and preventing subsequent cell adhesion. We have also made available a step-by-step experimental protocol for the fabrication and handling of microfluidic devices for investigating the interactions between antibiotics and individual bacteria (*Cama and Pagliara, 2021*).

## Imaging single-cell drug accumulation dynamics

An overnight culture was prepared as described above and typically displayed an optical density at 595 nm ($OD_{595}$) around 5. A 50 mL aliquot of the overnight culture above was centrifuged for 5 min at 4000 rpm and 37 °C. The supernatant was filtered twice (Medical Millex-GS Filter, 0.22 μm, Millipore Corp.) to remove bacterial debris from the solution and used to resuspend the bacteria in their spent LB to an $OD_{600}$ of 75. A 2 μL aliquot of this suspension was injected in each of the microfluidic networks above described and incubated at 37 °C. The high bacterial concentration favours bacteria entering the narrow lateral channels from the main microchamber of the mother machine (*Bamford et al., 2017*). We found that an incubation time between 5 and 20 min allowed filling of the lateral channels with, typically, between one and three bacteria per channel. Shorter incubation times were required for motile or small bacteria, such as *P. aeruginosa* and *S. aureus*, respectively. An average of 80% of lateral channels of the mother machine device were filled with bacteria. The microfluidic device was completed by the integration of fluorinated ethylene propylene tubing (1/32"×0.008"). The inlet tubing was connected to the inlet reservoir, which was connected to a computerised pressure-based flow control system (MFCS-4C, Fluigent). This instrumentation was controlled by MAESFLO software (Fluigent). At the end of the 20 min incubation period, the chip was mounted on an inverted microscope (IX73 Olympus, Tokyo, Japan) and the bacteria remaining in the main microchamber of the mother machine were washed into the outlet tubing and into the waste reservoir by flowing LB at 300 μL h$^{-1}$ for 8 min and then at 100 μL h$^{-1}$ for 2 h. Bright-field images were acquired every 20 min during this 2 hr period of growth in LB. Images were collected via a 60×, 1.2 N.A. objective (UPLSAPO60XW, Olympus) and a sCMOS camera (Zyla 4.2, Andor, Belfast, UK). The region of interest of the camera was adjusted to visualise 23 lateral channels per image and images of 10 different areas of the microfluidic device were acquired at each time point in order to collect data from at least 100 individual bacteria per experiment. The device was moved by two automated stages (M-545.USC and P-545.3C7, Physik Instrumente, Karlsruhe, Germany, for coarse and fine movements, respectively). After this initial 2 hr growth period in LB, the microfluidic environment was changed by flowing minimal medium M9 (unless otherwise stated) with each of the NBD (unless otherwise stated) fluorescent antibiotic derivatives at a concentration of 46 μg mL$^{-1}$ (unless otherwise stated, also unlabelled ciprofloxacin was delivered at 200 μg mL$^{-1}$) at 300 μL h$^{-1}$ for 8 min and then at 100 μL h$^{-1}$ for 4 h. During this 4 hr period of exposure to the fluorescent antibiotic derivative in use, upon acquiring each bright-field image the microscope was switched to fluorescent mode and FITC filter using Labview. A fluorescence image was acquired by exposing the bacteria for 0.03 s to the blue excitation band of a broad-spectrum LED (CoolLED pE300white, maximal power =200 mW Andover, UK) at 20% of its intensity (with a power associated with the beam light of 8 mW at the sample plane). In the case of unlabelled ciprofloxacin the UV excitation band of such LED was used at 100% of its intensity. These parameters were adjusted in order to maximise the signal to noise ratio. Bright-field and fluorescence imaging during this period was carried out every 5 min. The entire assay was carried out at 37 °C in an environmental chamber (Solent Scientific, Portsmouth, UK) surrounding the microscope and microfluidics equipment.

## Image and data analysis

Images were processed using ImageJ software as previously described (*Łapińska et al., 2019*; *Blaskovich et al., 2019*; *Smith et al., 2019*), tracking each individual bacterium throughout the initial 2 hr period of growth and the following 4 hr period treatment with each fluorescent antibiotic derivative. Briefly, during the initial 2 hr growth in LB, a rectangle was drawn around each bacterium in each bright-field image at every time point, obtaining its width, length, and relative position in the hosting microfluidic channel. Each bacterium's average elongation rate was calculated as the average of the ratios of the differences in bacterial length over the lapse of time between two consecutive time points. During the following 4 hr incubation in the presence of the fluorescent antibiotic derivative, a rectangle was drawn around each bacterium in each bright-field image at every time point, obtaining its width, length, and relative position in the hosting microfluidic channel. The same rectangle was then used in the corresponding fluorescence image to measure the mean fluorescence intensity for each bacterium that is the total fluorescence of the bacterium normalised by cell size (i.e. the area covered by each bacterium in our 2D images), to account for variations in antibiotic accumulation due to the cell cycle (*Taniguchi et al., 2010*). The same rectangle was then moved to the closest microfluidic channel that did not host any bacteria in order to measure the background fluorescence due to the presence of extracellular fluorescent antibiotic derivative in the media. This mean background fluorescence value was subtracted from the bacterium's fluorescence value. Background subtracted values smaller than 20 a.u. were set to zero since this was the typical noise value in our background measurements. All data were then analysed and plotted using GraphPad Prism 8. Statistical significance was tested using either paired or unpaired, two-tailed, Welch's *t*-test. Pearson correlation, means, SD, coefficients of variation, and medians were also calculated using GraphPad Prism 8.

## Inferring single-cell kinetic parameters of antibiotic accumulation via mathematical modelling

We constructed a minimal model of antibiotic accumulation in order to infer key kinetic parameters quantifying the accumulation of each antibiotic. We modelled antibiotic accumulation using the following set of ordinary differential equations (ODEs):

$$\frac{dc(t)}{dt} = r(t) - d_c c(t)$$
$$\frac{dr(t)}{dt} = k_1 U(t - t_0) - d_r r(t) - k_2 c(t)$$

where $U(t - t_0)$ represents the dimensionless step function:

$$U(t - t_0) = \begin{cases} 0, & t < t_0 \\ 1, & \geq t_0 \end{cases}$$

Variable $c(t)$ represents the intracellular antibiotic concentration (in arbitrary units [a.u.] of fluorescence levels), and $r(t)$ (a.u. s$^{-1}$) describes the antibiotic uptake rate. With the first equation we described how antibiotic accumulation, $c(t)$, changes over time as a result of two processes: (i) drug-uptake, which proceeds at a time-varying rate, $r(t)$; and (ii) drug loss (efflux or antibiotic transformation), which we modelled as a first order reaction with rate constant $d_c$ (s$^{-1}$). With the second equation we described the dynamics of time-varying antibiotic uptake rate, $r(t)$. The uptake rate starts increasing with a characteristic time-delay (parameter $t_0$), parameter $k_1$ (a.u. s$^{-2}$) is the associated rate constant of this increase. We also assumed a linear dampening effect (with associated rate constant $d_r$ [s$^{-1}$]) to constrain the increase in uptake rate, which allowed us to recapitulate the measured saturation in antibiotic accumulation. In this model the maximum saturation is given by $F_{max} = \frac{k_1}{d_r d_c}$. Finally, we introduced an adaptive inhibitory term (rate constant $k_2$ [a.u. s$^{-2}$]) to describe the dip observed in some single-cell trajectories in Figure 1-figure supplement 2 and Figure 1-figure supplement 3, which we assumed due to the fact that the presence of drugs intracellularly inhibits further drug uptake. We note that in this model we did not make any a priori assumptions about the mechanisms underlying antibiotic accumulation but rather aimed to capture the dynamics of the measured accumulation data.

Model parameters were inferred from single-cell fluorescence time-traces (see Image and data analysis section) using the probabilistic programming language Stan through its python interface

pystan (*Carpenter et al., 2017*). Stan provides full Bayesian parameter inference for continuous-variable models using the No-U-Turn sampler, a variant of the Hamiltonian Monte Carlo method. All No-U-Turn parameters were set to default values except parameter adapt_delta, which was set to 0.999 to avoid divergent runs of the algorithm. For each single-cell fluorescence time-trace the algorithm produced four chains, each one consisting of 3000 warm-up iterations followed by 1000 sampling iterations, giving in total 4000 samples from the parameters' posterior distribution. For each parameter, the median of the sampled posterior is used for subsequent analysis. For parameter inference, model time was rescaled by the length of the time-trace T, i.e., $t' = \frac{t}{T}$ so that time runs between 0 and 1, and model parameters were reparameterised (and made dimensionless) according to the rules ($d_c' = d_c/d_r, d_r' = d_r T, k_1' = k_1/d_r, k_2' = k_2/d_r, t_0' = t_0/T$). The following diffuse priors were used for the dimensionless parameters, where $U(a, b)$ denotes the uniform distribution in the range $[a, b]$: $d_c' \sim U(0, 1)$ so that uptake rate dynamics are always faster than drug-accumulation dynamics, i.e., $d_c < d_r$ ; $\log_{10} d_r' \sim U(0, 3)$ constraining the timescale associated with $d_r$ to be shorter than the timescale of the experiment, i.e., $1/ d_r < T$; $\log_{10} k_1' \sim U(0, 3)$ and $\log_{10} k_2' \sim U(-3, 0)$, so that the parameter controlling adaptive inhibition is small enough and there is no oscillatory behaviour in the model, i.e., $k_2 < k_1$; $t_0' \sim U(0, 1)$, since the transformed time $t'$ runs from 0 to 1.

## Statistical classification of the accumulation of antibiotics

For each cell, the marginal posterior distributions of all model parameters ($t_0$, $k_1$, $k_2$, $d_r$, $d_c$) were summarised using the corresponding first ($Q_1$), second ($Q_2$), and third ($Q_3$) quantiles. For each classification task, a statistical model (classification decision tree) was developed for predicting the drug class for each cell using the summarised parameter posterior distributions as input. Depending on the classification task, either all 5 parameters were considered (5 × 3 = 15 predictors) or just parameters $t_0$ and $k_1$ (2 × 3 = 6 predictors). Statistical classification was performed using Matlab (method *fitctree*) and the results presented were obtained using 10-fold cross-validation.

## Numerical simulations of antibiotic diffusion in microfluidic channels

Using COMSOL (version 5.5) with the Chemical Reaction Engineering module, we created a physical 2D model of the bacterial hosting microfluidic channel (with a length and width of 20 µm and 1 µm, respectively) with five bacteria (each with a length and width of 2 µm and 0.8 µm, respectively) arranged in the centre of the channel. We kept constant at $7\times10^4$ mol m$^{-3}$ (i.e. 46 µg mL$^{-1}$) the extracellular antibiotic concentration at the open end of the channel because in our experiments we continuously supplied antibiotics via the main channel of the device. We used a diffusion coefficient of $2.5\times10^{-10}$ m$^2$ s$^{-1}$ (i.e. the average of the diffusion coefficients estimated for polymyxin B-NBD and roxithromycin-NBD by using the Stokes-Einstein equation and the molecular weights reported in *Appendix 1—table 1*). Antibiotic binding to the bacterial surface was modelled with an absorption rate of 0.2 or 0.002 mol m$^{-2}$ s$^{-1}$ (i.e. this simple model does not intend to recapitulate the complex molecular transport across the bacterial double membrane), that linearly decreased to zero if the antibiotic concentration in the proximity of the bacterial surface was below 10% of the extracellular antibiotic concentration at the open end of the channel. Absorption at the surface of each bacterium stopped when the level of antibiotic concentration at the bacterial surface reached a value of 100 mol m$^{-2}$. After performing these simulations with a range of maximal concentrations from 10 to 1000 mol m$^{-2}$, the value 100 mol m$^{-2}$ was chosen empirically since it best matched the experimental observation that bacteria without screens reached saturation levels of intracellular polymyxin B fluorescence within minutes. Finally, when we run our simulations with an absorption rate value of 0.02 mol m$^{-2}$ s$^{-1}$ (i.e. intermediate between 0.2 and 0.002 mol m$^{-2}$ s$^{-1}$ values above), we obtained that only the simulated intracellular concentration of the most screened bacterium was affected by the presence of other bacteria.

## Acknowledgements

U.L. was supported through a BBSRC responsive mode grant (BB/V008021/1), an MRC Proximity to Discovery EXCITEME2 grant (MCPC17189) and an award from the Gordon and Betty Moore Foundation Marine Microbiology Initiative (GBMF5514). M.V. and K.T.A. gratefully acknowledge financial support from the EPSRC via grant EP/T017856/1. K.K.L was supported via a Living Systems Institute PhD studentship. A.C. and B.T. were supported via an EPSRC DTP PhD studentship (EP/

M506527/1). M.R.L.S. was supported by an Australian Postgraduate Award and an Institute for Molecular Biosciences Research Advancement Award. B.Z was supported by a CSC scholarship. M.A.T.B. was supported in part by Wellcome Trust Strategic Grant WT1104797/Z/14/Z and NHMRC Development grant APP1113719. This work was further supported by a Royal Society Research Grant (RG180007) awarded to S.P, a QUEX Initiator grant awarded to S.P, K.T.A. and M.A.T.B., an NHMRC Ideas grant (2004367) awarded to M.A.T.B, and a GW4 Initiator award to M.V., K.T.A and S.P. S.P.'s work in this area is also supported by a Marie Skłodowska-Curie project SINGEK (H2020-MSCA-ITN-2015–675752).

## Additional information

### Funding

| Funder | Grant reference number | Author |
|---|---|---|
| Biotechnology and Biological Sciences Research Council | BB/V008021/1 | Urszula Łapińska Stefano Pagliara |
| Medical Research Council | MCPC17189 | Urszula Łapińska Stefano Pagliara |
| Gordon and Betty Moore Foundation | GBMF5514 | Stefano Pagliara |
| Engineering and Physical Sciences Research Council | EP/T017856/1 | Margaritis Voliotis Krasimira Tsaneva-Atanasova |
| Engineering and Physical Sciences Research Council | EP/M506527/1 | Adrian Campey |
| Wellcome Trust | WT1104797/Z/14/Z | Mark AT Blaskovich |
| Royal Society | RG180007 | Stefano Pagliara |
| H2020 Marie Skłodowska-Curie Actions | H2020-MSCA-ITN-2015-675752 | Stefano Pagliara |

The funders had no role in study design, data collection and interpretation, or the decision to submit the work for publication. For the purpose of Open Access, the authors have applied a CC BY public copyright license to any Author Accepted Manuscript version arising from this submission.

### Author contributions

Urszula Łapińska, Conceptualization, Formal analysis, Funding acquisition, Investigation, Methodology, Supervision, Writing - original draft, Writing – review and editing; Margaritis Voliotis, Formal analysis, Methodology, Software, Writing – review and editing; Ka Kiu Lee, Adrian Campey, Data curation, Investigation, Methodology, Writing – review and editing; M Rhia L Stone, Wanida Phetsang, Bing Zhang, Methodology, Writing – review and editing; Brandon Tuck, Formal analysis, Software, Writing – review and editing; Krasimira Tsaneva-Atanasova, Formal analysis, Funding acquisition, Writing – review and editing; Mark AT Blaskovich, Data curation, Funding acquisition, Writing – review and editing; Stefano Pagliara, Conceptualization, Formal analysis, Funding acquisition, Project administration, Supervision, Writing - original draft, Writing – review and editing

### Author ORCIDs

Urszula Łapińska http://orcid.org/0000-0003-3593-9248
Margaritis Voliotis http://orcid.org/0000-0001-6488-7198
Krasimira Tsaneva-Atanasova http://orcid.org/0000-0002-6294-7051
Stefano Pagliara http://orcid.org/0000-0001-9796-1956

### Decision letter and Author response

Decision letter https://doi.org/10.7554/eLife.74062.sa1
Author response https://doi.org/10.7554/eLife.74062.sa2

## Additional files

### Supplementary files
• Transparent reporting form

### Data availability
All data acquired for this study are presented within the manuscript, the supplementary information and the source data files.

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

# Appendix 1

**Appendix 1—table 1.** List of fluorescent antibiotic derivatives (obtained by linking the parental antibiotic to nitrobenzoxadiazole, NBD, see Methods), the bacterial compartment where their target is located, their molecular weight (MW) after linkage to NBD, their partition coefficient (logP), their measured minimum inhibitory concentration (MIC) against *E. coli* BW25113, and the fold-change compared to the MIC measured for each corresponding parental antibiotic (see Methods).
MIC data were collated from biological triplicate.

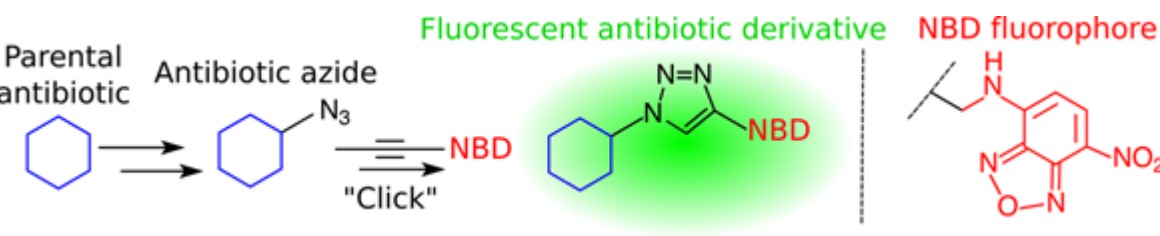

| Antibiotic probe | Compartment | MW (g/mol) | logP | MIC (µg/mL) | Fold change |
|---|---|---|---|---|---|
| Polymyxin B-NBD | Membrane | 1,449 | –2.5 | 1 | 1 |
| Octapeptin-NBD | Membrane | 1,304 | –0.4 | 4 | 1 |
| Tachyplesin-NBD | Membrane | 2,523 | –2.7 | 1 | 1 |
| Vancomycin-NBD | Cell wall | 1,650 | –2.6 | >192 | 1 |
| Linezolid-NBD | Cytoplasm | 638 | 0.7 | 134 | 1 |
| Roxithromycin-NBD | Cytoplasm | 1,064 | 3.1 | 192 | 3 |
| Ciprofloxacin-NBD | Cytoplasm | 633 | –1.1 | 8 | 256 |
| Trimethoprim-NBD | Cytoplasm | 577 | 0.9 | 64 | 64 |

**Appendix 1—table 2.** Pearson correlation coefficients and significance of the correlation between $t_0$ and $k_1$, $t_0$ and $F_{max}$ and $k_1$ and $F_{max}$ for the accumulation in single *E. coli* of all the fluorescent antibiotic derivatives investigated (apart from vancomycin) in individual *E. coli*.
Data from *Figure 1—figure supplement 13* were used for these statistical comparisons. ****: p-value<0.0001, ***: p-value <0.001, **: p-value<0.01, *: p-value<0.05, ns: not significant, p-value>0.05.

| Antibiotics | t0 vs k1 | t0 vs Fmax | k1 vs Fmax |
|---|---|---|---|
| | \multicolumn Pearson correlation coefficients and significance | | |
| Polymyxin B | –0,51, **** | –0,54, **** | 0,56, **** |
| Octapeptin | –0,46, **** | –0,61, **** | 0,20, ns |
| Tachyplesin | –0,13, ns | –0,10, ns | –0,01, ns |
| Linezolid | 0.03, ns | –0,21, ** | 0,05, ns |
| Ciprofloxacin | –0,12, ns | –0,11, ns | 0,29, *** |
| Trimethoprim | 0,06, ns | –0,32, **** | 0,11, ns |
| Roxithromycin | –0,22, *** | –0,10, ns | 0,41, **** |
| All antibiotics | –0,40, **** | –0,27, **** | 0,65, **** |

**Appendix 1—table 3.** List of genes encoding outer membrane proteins (i.e. porins) and efflux pumps compiled using EcoCyc as previously reported (***Kortright et al., 2020***), alongside their transcript reads after a 2 hr growth period in lysogeny broth (LB) (i.e. the time point at which antibiotic treatment starts in our microfluidic experiments) measured via RNA-sequencing as previously reported (***Smith et al., 2018***). Note that it has been reported that permeability of solutes through OmpA (with the most highly expressed transcripts) is a 100-fold lower compared to that

through OmpC (**Sugawara and Nikaido, 1992**) (with the second most highly expressed transcripts), hence we decided to investigate the role played by OmpC in the heterogeneity in the intracellular accumulation of roxithromycin (**Figure 4E**).

| Membrane genes | Transcript reads | Membrane genes | Transcript reads | Membrane genes | Transcript reads | Membrane genes | Transcript reads |
|---|---|---|---|---|---|---|---|
| ompA | 60,955 | mltA | 398 | acrZ | 47 | yaiO | 6 |
| ompC | 57,458 | yncD | 388 | yfiB | 39 | cusB | 6 |
| ompX | 19,210 | lolB | 345 | cusA | 39 | yehB | 5 |
| lptD | 10,977 | nlpD | 326 | macb | 36 | bglH | 5 |
| tolC | 4722 | mdtK | 312 | yhcD | 33 | wza | 5 |
| fhuA | 4360 | yiaD | 292 | fimD | 31 | blc | 5 |
| bamA | 4237 | nplE | 291 | acrF | 30 | acrE | 5 |
| acrB | 4044 | fepA | 289 | pgaA | 29 | yfgH | 4 |
| bamB | 3796 | yraP | 256 | mdtL | 28 | nanC | 4 |
| ompF | 3650 | emtA | 252 | mdtG | 28 | yqhH | 4 |
| slyB | 3516 | ydiY | 241 | mdtF | 27 | phoE | 4 |
| nlpI | 3367 | tamA | 236 | yfaL | 25 | mdtQ | 3 |
| fadL | 2612 | yjgL | 222 | gfcD | 24 | yliI | 3 |
| ompT | 2601 | mdfA | 220 | gspD | 23 | ompN | 3 |
| mipA | 2289 | ynfB | 220 | yraJ | 22 | mdtO | 3 |
| mltD | 2045 | ypjA | 220 | gfcE | 22 | cusC | 2 |
| fecA | 2009 | pgpB | 193 | flgG | 22 | cusF | 2 |
| tsx | 1971 | mltC | 187 | mdtJ | 21 | mdtP | 2 |
| pal | 1945 | mdtC | 166 | mdtD | 19 | yfeN | 2 |
| skp | 1553 | lpoB | 155 | ydeT | 17 | mdtN | 2 |
| bamD | 1544 | macA | 153 | slp | 16 | csgF | 2 |
| acrA | 1505 | loiP | 137 | yceK | 16 | yjbF | 1 |
| mepS | 1303 | mltF | 134 | mdtI | 13 | csgB | 1 |
| lpp | 1168 | yaiW | 131 | chiP | 12 | envY | 1 |
| borD | 1167 | bhsA | 119 | pagP | 11 | ybgQ | 1 |
| nmpC | 1146 | pqiC | 114 | yedS | 11 | acrS | 1 |
| cirA | 1127 | rsxG | 107 | yjbH | 10 | uidC | 1 |
| bamC | 1123 | rcsF | 105 | rhsD | 9 | csgE | 0 |
| ygiB | 1115 | yfaZ | 101 | elfC | 9 | ompL | 0 |
| flu | 1064 | cusR | 99 | rhsB | 9 | ompG | 0 |
| lptA | 1052 | nfrA | 98 | yfcU | 8 | rzoD | 0 |
| mlaA | 1042 | cusS | 92 | lamB | 8 | rzoR | 0 |
| ybhC | 1021 | acrR | 85 | pgaB | 8 | yddL | 0 |
| lptE | 924 | yghG | 83 | sfmD | 8 | appX | ND |
| bamE | 732 | fhuE | 81 | htrE | 8 | bcsC | ND |
| rlpA | 637 | amiD | 80 | mdtH | 7 | epcC | ND |
| lpoA | 620 | yddB | 76 | yiaT | 7 | qseG | ND |

*Appendix 1—table 3 Continued on next page*

*Appendix 1—table 3 Continued*

| Membrane genes | Transcript reads | Membrane genes | Transcript reads | Membrane genes | Transcript reads | Membrane genes | Transcript reads |
|---|---|---|---|---|---|---|---|
| *fiu* | 607 | *acrD* | 72 | *mliC* | 7 | *ychO* | ND |
| *btuB* | 489 | *ecnB* | 69 | *mdtE* | 7 | *ypjB* | ND |
| *tamB* | 408 | *mdtB* | 64 | *flgH* | 7 | *yzcX* | ND |
| *mltB* | 406 | *mdtA* | 54 | *csgG* | 6 | | |
| *pldA* | 404 | *ecnA* | 51 | *hofQ* | 6 | | |
| *ppk* | 404 | *mdtM* | 49 | *ompW* | 6 | | |

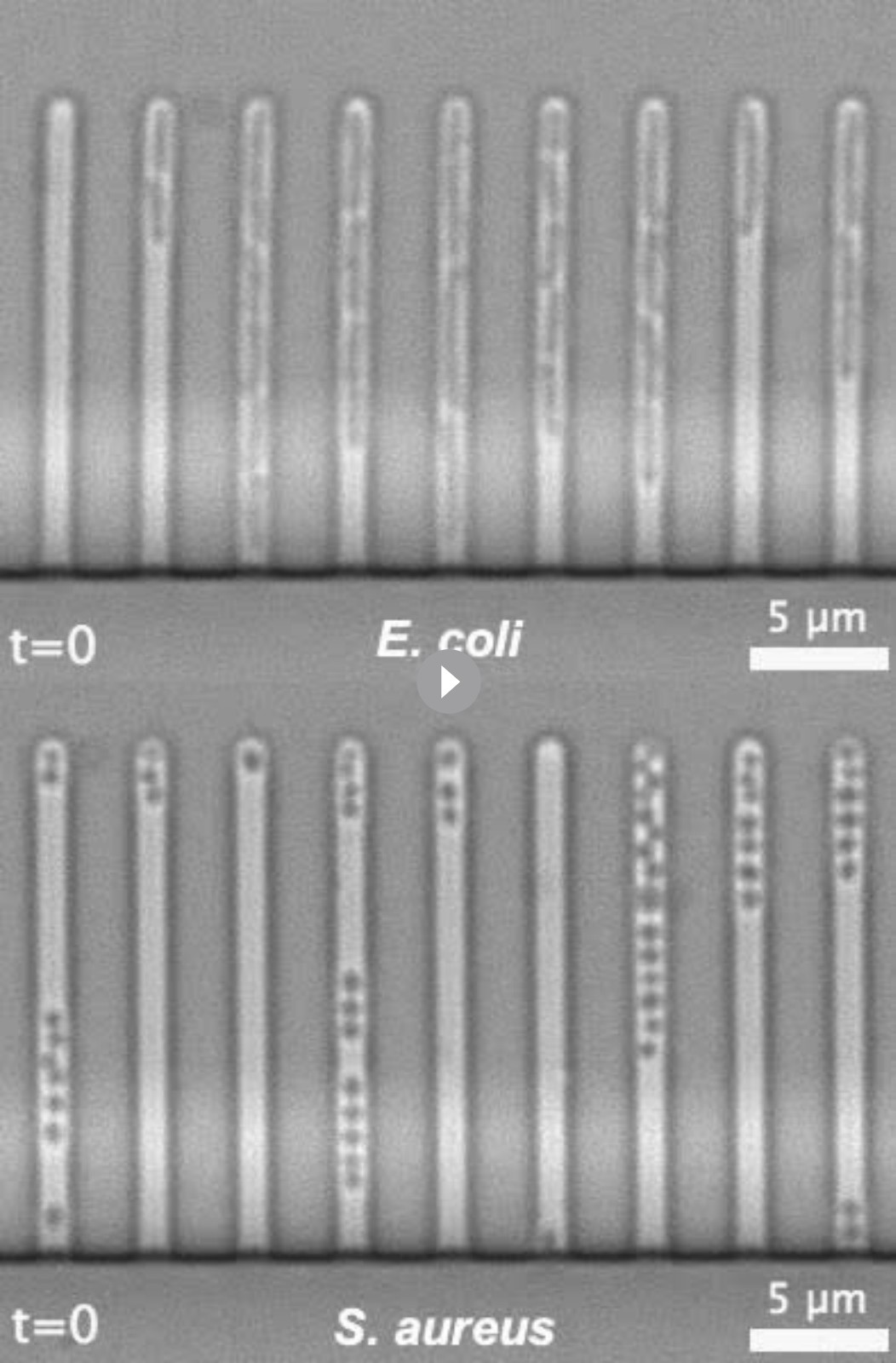

**Appendix 1—video 1.** Comparison of roxithromycin accumulation in gram-negative and gram-positive bacteria. Time-lapse microscopy displaying the accumulation of roxithromycin-NDB in individual *E. coli* (top) and *S. aureus* bacteria (bottom). Roxithromycin-NDB was added to the microfluidic device at t=0 at an extracellular concentration of 46 μg mL⁻¹.

https://elifesciences.org/articles/74062/figures#video1

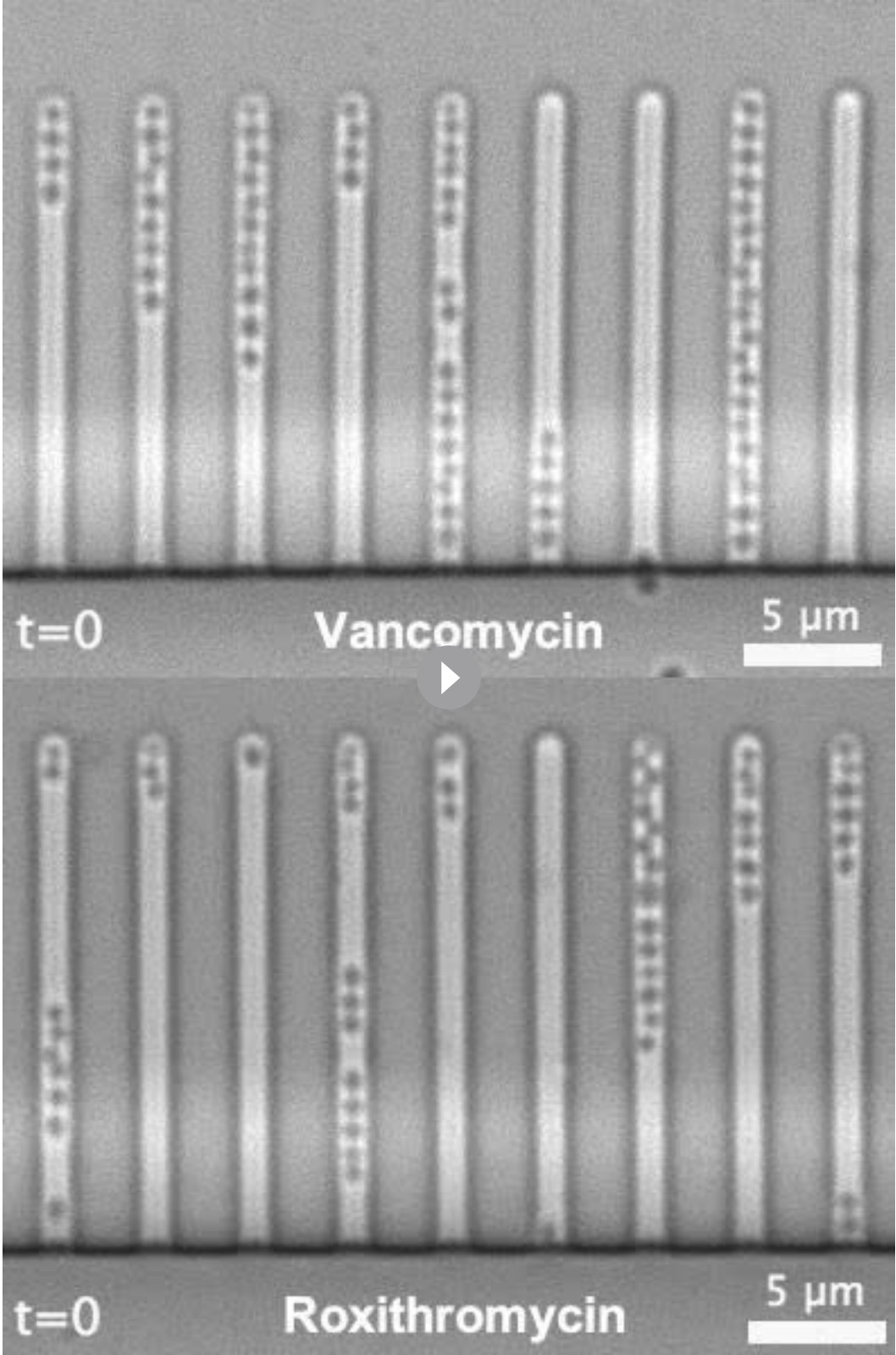

**Appendix 1—video 2.** Comparison of vancomycin and roxithromycin accumulation in gram-positive bacteria. Time-lapse microscopy displaying the accumulation of vancomycin-NDB (top) and roxithromycin-NDB (bottom) in individual *S. aureus* bacteria. Both fluorescent antibiotic derivatives were added to the microfluidic device at t=0 at an extracellular concentration of 46 µg mL$^{-1}$.
https://elifesciences.org/articles/74062/figures#video2

