## [Editor Report]

This study addresses mechanisms by which bacteria are able to survive and evade killing by antibiotics. Using fluorescent versions of antibiotics it studies whether entry/efflux of the drug itself is a significant contributor to the observed variability of antibiotic activity. This study will be of interest to microbiologists and clinicians for the design of better antibiotic therapies and improves our understanding of the relationships between drug uptake, bacterial growth, and drug efficacy.

---

## [Decision Letter]

**Decision letter after peer review**

Thank you for submitting your article ‘Fast bacterial growth reduces antibiotic accumulation and efficacy’ for consideration by *eLife*. Your article has been reviewed by 2 peer reviewers, and the evaluation has been overseen by a Reviewing Editor and Gisela Storz as the Senior Editor. The reviewers have opted to remain anonymous.

Essential revisions

1) The authors report a predrug growth rate variability of more than an order of magnitude (Figure 5A-C), which also seems to vary drastically between experiments (compare x-axis range in Figure 5A-C with Figure 5E-F). This variation is considerably larger than what has been reported by other groups using very similar methods and conditions (e.g. Figure 2 in https://doi.org/10.1016/j.cub.2010.04.045 shows a spread of only a few percent in division time). This large spread possibly occurs because the cells have not yet reached steady state growth conditions before antibiotic addition. The 2 h growth period in LB before antibiotic addition indeed seems insufficient for reaching a new steady state of exponential growth; at the end of this period, large parts of the population are likely still in lag phase or stressed from loading into the microfluidic device. In any case, the state of the cells at drug addition and its reproducibility is not clear.

2) The authors claim that growth-dependent drug sensitivity for roxithromycin is because of growth-dependent transport rates. This conclusion seems insufficiently supported by data.

– Roxithromycin is the only drug that shows a negative correlation between growth rate and drug increase rate—and it is also the drug with the slowest net drug increase. Indeed, the changes in drug concentration are slow compared to the division time. Therefore, dilution via growth might be sufficient to explain the correlation without the need for any more detailed molecular mechanism. Importantly, does a TolC knockout reduce the growth rate in the presence of roxithromycin? If so, (how) can one distinguish a general growth rate effect from a specific TolC effect?

– If the authors can indeed clearly show that the effect is TolC-specific, it would be interesting to also measure the difference in drug uptake kinetics of an ompC mutant. OmpC can still play a role in drug uptake kinetics, even if its expression does not correlate with growth rate.

3) It is not clear why correlations of tolC and ompC expression with the growth before drug addition are analyzed (Figure 5). Since these proteins can affect drug uptake and efflux, it seems more relevant to directly test their correlation with the parameters quantifying these phenomena. To rule out various well-known non-specific effects of GFP reporters observed at different growth rates (e.g. copy number variation of the reporter gene, dependence of maturation time and other properties of GFP on the cytosolic milieu etc.), it would be important to perform controls using several unrelated promoters to corroborate the relevance of the correlations in Figure 5D-F.

*Reviewer #1 (Recommendations for the authors)*

There have been several studies that have visualised drug accumulation in bacteria at the single-cell level (Cinquin et al., – PMID26656111; Reuter et al., PMID32761242). However, in this study, the authors have done a more thorough job by looking at different classes of antibiotics, against both gram-negative and gram-positive bacteria. They also directly address the functional consequences of decreased accumulation and show that at least for one of the compounds survival was not linked with slower growth rates.

Overall, the study is well designed and the authors have carried out exhaustive characterisation of the different compounds. However, based on their results, what strongly comes across is that there are no unifying principles. Accumulation properties and correlations are unique to each antibiotic and behaviors cannot be generalised even across antibiotics targeting the same sub-cellular space or target. Therefore, some of the conclusions needs to be toned down accordingly.

Specific comments

– Ln 103-104. Table S1, the effect of the fluorescent tag was not neutral across different compounds, some of them exhibited a fold-shift in MIC of 3-256 fold, which are not-insignificant.

– It is clear that for the results to be comparable across different compounds, the authors used the same concentration of compound (46 ug/ml). However, this dose would represent 46x MIC for compounds such as polymyxin B and 0.23x MIC for roxithromycin. Considering that the different fold MICs have hugely differing impact on growth and cellular processes, this would probably translate to differences in accumulation properties depending upon the fold MIC for that particular compound. In Figure S9B, the authors are probably trying to address this issue by using 192 and 46 ug/ml of roxithromycin, but these would represent 1x and 0.23x MICs. It would be more relevant to use 20-40x MIC concentrations of roxithromycin or if solubility is an issue, to use Polymyxin at 1x and 46x concentrations.

– Ln 173-183. It could be useful to provide MIC values of roxithromycin-NBD and vancomycin-NBD for S. aureus and ciprofloxacin-NBD for *P. aeruginosa* and *B. cenocepacia*.

– Ln 213, – Linezolid despite being an intracellular targeting drug, seems to exhibit accumulation kinetics similar to membrane targeting compounds. Maybe the authors can discuss about this in more detail.

– Figure 2. I don’t see the value of this figure. The data shown in panels A and B are quite scattered and the correlations are not so obvious.

– Data in Figure 3C and 5A are plotting correlations of the same two parameters, but the values shown are different.

– Ln 448-451 – The authors generated a tolC reporter strain to address the link between kinetic parameter t0 and tolC expression. However, they do not show this correlation but based on positive correlations between t0 and elongation rate and between GFP and elongation rate, conclude that it is in line with their hypothesis. This is erroneous. This would need comparing of GFP levels with t0 and correlating directly. Similar logic was used for in subsequent paragraphs for panels 5E and 5F.

*Reviewer #2 (Recommendations for the authors)*

– To check whether the dye has an effect on the drug activity, it would be informative to compare single cell growth variability with labelled and unlabeled compounds.

– Background fluorescence is analyzed in empty channels. Another relevant control to rule out that autofluorescence affects the results is to measure fluorescence in cells that are treated with the non-fluorescent versions of the antibiotics.

– Quite a few claims and implications seem exaggerated

– Line 569: the authors claim to mechanistically understand why some drugs are more effective in biofilms than others. Whilst their Figure 4 is interesting, and seems in line with the cited references, it remains unclear why some drugs show a trend as a function of position whereas others don’t.

– Identifying the target of antibiotics via single cell drug uptake kinetic measurements seems a bit farfetched as an application. First, following the data provided by the authors, one could also classify the drugs using (more high throughput) bulk measurements. Second, the authors should compare their technique in the context of current approaches used to classify drugs to clarify its advantages and disadvantages. Lastly, in line 292 it says that this technique can help ‘phenotyping bacterial populations’. What do the authors mean here? Microfluidic techniques to rapidly phenotype drug sensitivity have already been proposed without using fluorescent antibiotics – it is not clear how much fluorescence would add here.

– Line 491 suggests that reducing heterogeneity is the main goal of any treatment. However, the more established goal is to inhibit the growth of as many cells as possible. This can be achieved by keeping the same heterogeneity but simply reducing the mean.

– In line 515 ‘needs to be’ seems exaggerated; similarly ‘major rethink’ in line 543.

– The term ‘phenotypically engineer’ is used both in the abstract and in line 581. This term is unclear for adding an extra compound.

– Figure 3a shows four exemplary curves out of a large data set. It would help to rigorously analyze the reduction in growth rate after the supposed lag time for the entire dataset.

– Line 587: The ‘biases’ mentioned here need to be explained. As is, this claim is unclear without digging into literature.

[Editors’ note: further revisions were suggested prior to acceptance, as described below.]

Thank you for resubmitting your work entitled ‘Fast bacterial growth reduces antibiotic accumulation and efficacy’ for further consideration by *eLife*. Your revised article has been evaluated by Gisela Storz (Senior Editor) and a Reviewing Editor.

The manuscript has been improved but there are some remaining issues that need to be addressed, as outlined below:

Specifically, the extent of heterogeneity in elongation rates disagrees significantly from prior work, and the current version still does not address prior concerns from Reviewer 2. The robustness of the findings of this study appear to hinge on these concerns, so despite our enthusiasm for the principle, the distribution of growth rates must be well accounted for.

*Reviewer #1 (Recommendations for the authors)*

The authors have addressed and made modifications to the revised text and have incorporated these new data in the revised manuscript.

I am satisfied with the current version of the manuscript.

Only one important edit. Figure 4D-F needs to be updated and reflect the new content being described in the text and figure legend. The correct figure was shown in the response to the reviewers letter but the figure in the main file is still the previous version.

*Reviewer #2 (Recommendations for the authors)*

The manuscript has certainly improved, especially due to the inclusion of new control experiments and some simulations. However, some of my concerns remain.

1. I am not convinced that the intra-experiment heterogeneity in elongation rate is similar to that reported in literature as claimed in the response. For example, the elongation rates shown in Figure 2 in Wang et al., are approximately normally distributed with almost all data falling between 0.03/min and 0.06/min; no elongation rates are anywhere near zero. Similarly, in Baltekin et al., Figure 2F (left panel) virtually all (normalised) growth rates fall between 0.5 and 1.25 with a handful of lower values and no values near zero. In both papers, the numbers of cells analyzed are orders of magnitude larger, making outliers more likely. In the preset work, while fewer cells were analyzed, the elongation rates are spread from 0 to 15 μm/h with many values near zero (Figure 4). Therefore, I cannot follow the authors' reasoning here: the variability in elongation rates they observed is clearly much greater than what has been reported in literature. This is problematic for the reasons described in my first report.

2. The authors argue that dilution via growth is unlikely to play a role because they rarely observed cell division during their experiment. However, the two fundamental timescales here are the doubling time and the drug uptake time. The authors state that the doubling time is 75 min = 4500 seconds, while uptake of roxithromycin takes more than 5000 seconds (Figure 1). Therefore, regardless of whether the experiment was performed long enough to observe cell division, drug uptake is slow compared to dilution via growth of cell volume (which happens continuously irrespective of cell divisions). This effect therefore inevitably contributes to the net drug uptake kinetics. One could try to quantitatively correct for this effect; this would be important to rule out that it largely explains the observed correlation between growth rate and net uptake.

---

## [Author Response]

Essential revisions:1) The authors report a pre-drug growth rate variability of more than an order of magnitude (Figure 5A-C), which also seems to vary drastically between experiments (compare x-axis range in Figure 5A-C with Figure 5E-F). This variation is considerably larger than what has been reported by other groups using very similar methods and conditions (e.g. Figure 2 in https://doi.org/10.1016/j.cub.2010.04.045 shows a spread of only a few percent in division time). This large spread possibly occurs because the cells have not yet reached steady state growth conditions before antibiotic addition. The 2 h growth period in LB before antibiotic addition indeed seems insufficient for reaching a new steady state of exponential growth; at the end of this period, large parts of the population are likely still in lag phase or stressed from loading into the microfluidic device. In any case, the state of the cells at drug addition and its reproducibility is not clear.

First, we would like to thank the reviewer for noticing the discrepancy between the elongation rates in Figures 5A-C and Figures 5D-F of the submitted manuscript (now Figures 4A-C and Figures 4DF). This was due to the use of an erroneous pixel to µm correction factor that we have now rectified as reported in the amended Figure 4 (please note that we have then further amended this figure according to essential revision 3 below). The elongation rates averaged over three independent experiments performed with the parental, *tolC*, *ompC* and *rrnB* strains were (5.3±1.2) µm hr^-1^, (4.9±0.4) µm hr^-1^, (5.4±0.5) µm hr^-1^ and (4.2±0.7) µm hr^-1^, respectively. This inter-experimental variability is in agreement with the one reported in the landmark paper by Wang et al*.,* suggested by the reviewer and cited in our manuscript (https://doi.org/10.1016/j.cub.2010.04.045). In fact, Wang et al*.,* reported doubling times ranging between 0.04 and 0.05 divisions min^-1^ across the different strains they examined. Moreover, we have now run new experiments to further verify that the elongation rate does not substantially vary after 2 h post-addition in the microfluidic device (new Figure 1—figure supplement 1). This data is in accordance with our previous bulk data showing that *E. coli* is in exponential phase between 2 and 5 h after passage in fresh medium (see for example Figure 3A in Smith et al*., Front Microbiol* 9, 1739, 2018). In terms of intra-experimental variability the coefficients of variations for the elongation rate of the parental, *tolC*, *ompC* and *rrnB* strains were 55, 41, 36, and 49%, respectively, in line with previously reported single-cell growth rate variabilities both in the absence and in the presence of antibiotics (see for example Figures 2 and 3 in Baltekin et al*., PNAS* 114, 9170, 2017).

These new data are discussed on lines 97-101 and 436-439 of our revised manuscript with track changes (please note that line numbers might be different in the clean copy of the manuscript because we noticed that some parts of the manuscript moved around during the pdf building process on *eLife* portal) and in the new Figures 4 and Figure 1—figure supplement 1.

2) The authors claim that growth-dependent drug sensitivity for roxithromycin is because of growth-dependent transport rates. This conclusion seems insufficiently supported by data.– Roxithromycin is the only drug that shows a negative correlation between growth rate and drug increase rate – and it is also the drug with the slowest net drug increase. Indeed, the changes in drug concentration are slow compared to the division time. Therefore, dilution via growth might be sufficient to explain the correlation without the need for any more detailed molecular mechanism. Importantly, does a TolC knockout reduce the growth rate in the presence of roxithromycin? If so, (how) can one distinguish a general growth rate effect from a specific TolC effect?– If the authors can indeed clearly show that the effect is TolC-specific, it would be interesting to also measure the difference in drug uptake kinetics of an ompC mutant. OmpC can still play a role in drug uptake kinetics, even if its expression does not correlate with growth rate.

We thank the reviewer for giving us the opportunity to expand on this important point. Indeed in our submitted manuscript we had very briefly acknowledged the possibility that dilution of the intracellular concentration via bacterial doubling could account for reduced intracellular drug accumulation. To minimise this effect we investigated the accumulation of all our fluorescent antibiotic derivatives using minimal medium M9 as the drug milieu (please see lines 149-151 of the submitted manuscript). As a consequence, during drug treatment bacterial divisions were rare events and thus the changes in drug concentration were not slow compared to the division time. In fact, during treatment with roxithromycin dissolved in M9 (data in Figure 1B of the submitted paper) none of the analysed bacteria underwent a full cell cycle from birth to division. Moreover, 15% of the bacteria analysed underwent one division during treatment but were born before drug treatment. The average doubling time of these bacteria was (75±28) min (new Figure 1—figure supplement 5A). In comparison, during treatment with roxithromycin dissolved in LB (data in Figure S9A of the submitted paper, now Figure 1—figure supplement 11A) 41% of the analysed bacteria underwent a full cell cycle from birth to division. Furthermore, when including bacteria that underwent one division during treatment but were born before drug treatment, the average doubling time ((29±9) min) was significantly shorter compared to that measured for treatment in M9 (****, Figure 1figure supplement 5). However, we found higher roxithromycin-NBD accumulation (Figure 1—figure supplement 11A) when LB was used as drug milieu. We also found similarly large heterogeneities when roxithromycin-NBD was dissolved in M9 or LB (CV in range of 84-372% and 51-428%, respectively). Taken together these data demonstrate that heterogeneity in roxithromycin accumulation cannot be explained via dilution due to bacterial doubling.

Moreover, when using the D*tolC* knockout and M9 as the drug milieu we found again that none of the analysed bacteria underwent a full cell cycle from birth to division and 13% of the bacteria analysed underwent one division during treatment but were born before drug treatment. The average doubling time of these bacteria was (80±30) minutes and the distribution of doubling times was not significantly different compared to the distribution of doubling times measured when using the parental strain and M9 as the drug milieu (p-value=0.51, Figure 1—figure supplement 5A). Therefore, the measured impact of the absence of *tolC* on roxithromycin accumulation (i.e. significantly shorter *t_0_* and larger *k_1_*) can not be ascribed to dilution via cell doubling. Finally, we have now carried out new experiments that demonstrate that, under the experimental conditions investigated, the outer membrane porin OmpC does not play a significant role in roxithromycin accumulation (Figure 5—figure supplement 1), thus corroborating our previous data obtained using an *ompC* fluorescent reporter strain.

These new data are discussed in the revised manuscript on lines 142-150, 185-192, 523-526, and in the new Figure 1—figure supplement 5 and Figure 5—figure supplement 1.

3) It is not clear why correlations of tolC and ompC expression with the growth before drug addition are analyzed (Figure 5). Since these proteins can affect drug uptake and efflux, it seems more relevant to directly test their correlation with the parameters quantifying these phenomena. To rule out various well-known non-specific effects of GFP reporters observed at different growth rates (e.g. copy number variation of the reporter gene, dependence of maturation time and other properties of GFP on the cytosolic milieu etc.), it would be important to perform controls using several unrelated promoters to corroborate the relevance of the correlations in Figure 5D-F.

We agree with the reviewer that it is more relevant to directly show the correlation between the expression of the investigated promoter and the kinetic parameters quantifying roxithromycin accumulation. Therefore, we have amended Figures 5D-F (now Figures 4D-F) accordingly. We also agree with the reviewer that it is important to perform independent experiments to further corroborate these findings. Indeed we have now shown that a deletion mutant lacking *tolC* displays shorter *t_0_* and larger *k_1_* compared to the parental strain (Figure 6 of the submitted paper, now Figure 5), whereas a deletion mutant lacking *ompC* accumulates roxithromycin in a similar fashion as the parental strain (please see new Figure 5—figure supplement 1). Moreover, following the reviewer suggestion we have further verified that there exist a significant negative correlation also between *F_max_* and the expression of *rrnH* (new Figure 4—figure supplement 3), a different *rrn* operon that is expressed at lower levels compared to *rrnB* (Maeda et al*., PLoS One* 10, e0144697, 2015).

These new data are discussed in the revised manuscript on lines 473-474, 479-482, 487-489, 491495, and in the new Figures 4 and Figure 4—figure supplement 3.

Reviewer #1 (Recommendations for the authors):There have been several studies that have visualised drug accumulation in bacteria at the single-cell level (Cinquin et al., – PMID26656111; Reuter et al., PMID32761242). However, in this study, the authors have done a more thorough job by looking at different classes of antibiotics, against both gram-negative and gram-positive bacteria. They also directly address the functional consequences of decreased accumulation and show that at least for one of the compounds survival was not linked with slower growth rates.Overall, the study is well designed and the authors have carried out exhaustive characterisation of the different compounds. However, based on their results, what strongly comes across is that there are no unifying principles. Accumulation properties and correlations are unique to each antibiotic and behaviors cannot be generalised even across antibiotics targeting the same sub-cellular space or target. Therefore, some of the conclusions needs to be toned down accordingly.

We would like to thank the reviewer for appreciating our novel approach to investigate bacterial heterogeneity in the response to antibiotics and for their constructive feedback. We had already cited the landmark work from the Pages' group using microspectroscopy to study drug accumulation at the single-cell level (references 44-46 in our revised manuscript) and we have now cited the work of Reuter et al., (reference 47 in our revised manuscript). Moreover, we have now toned down our conclusions following the comments of both reviewers, please see lines 75-77, 295-297, 346-347, 363, 416, 449, 470, 499-501, 548-551, 579-580, 604-606, 619-622 of our revised manuscript.

Specific comments:– Ln 103-104. Table S1, the effect of the fluorescent tag was not neutral across different compounds, some of them exhibited a fold-shift in MIC of 3-256 fold, which are not-insignificant.

We have now explicitly acknowledged that the fluorescent derivatives of polymyxin B, octapeptin, tachyplesin, vancomycin and linezolid maintained the antibiotic activity of the parent drug, whereas the fluorescent derivatives of roxithromycin, trimethoprim and ciprofloxacin displayed a 3-, 64- and 256-fold increase compared to the parent drug, respectively.

This new information is discussed on lines 88-92 of our revised manuscript.

– It is clear that for the results to be comparable across different compounds, the authors used the same concentration of compound (46 ug/ml). However, this dose would represent 46x MIC for compounds such as polymyxin B and 0.23 X MIC for roxithromycin. Considering that the different fold MICs have hugely differing impact on growth and cellular processes, this would probably translate to differences in accumulation properties depending upon the fold MIC for that particular compound. In Figure S9B, the authors are probably trying to address this issue by using 192 and 46 ug/ml of roxithromycin, but these would represent 1x and 0.23x MICs. It would be more relevant to use 20-40X MIC concentrations of roxithromycin or if solubility is an issue, to use Polymyxin at 1x and 46x concentrations.

We have now carried out new experiments using polymyxin B-NBD at the MIC (i.e. 1 µg ml^-1^) as suggested by the reviewer. As expected, we found that at this lower extracellular concentration, polymyxin B-NBD accumulated in individual *E. coli* more slowly (compared to a 46 µg ml^-1^ extracellular concentration). However, the heterogeneity in polymyxin B accumulation was similarly large at both concentrations with CVs in the range 9-250 and 11-212% in the case of 1 and 46 µg ml^-1^ extracellular concentration, respectively.

These new data are discussed in the revised manuscript on lines 193-195 and in the new Figure 1 figure supplement 11C.

– Ln 173-183. It could be useful to provide MIC values of roxithromycin-NBD and vancomycin-NBD for S. aureus and ciprofloxacin-NBD for P. aeruginosa and B. cenocepacia.

MIC values of roxithromycin-NBD and vancomycin-NBD for *S. aureus* (1 and 0.5 µg ml^-1^, respectively) and ciprofloxacin-NBD for *P. aeruginosa* and *B. cenocepacia* (32 µg ml^-1^ for both species) have now been provided in the captions of Figure 1—figure supplements 8, 9, and 10.

– Ln 213, – Linezolid despite being an intracellular targeting drug, seems to exhibit accumulation kinetics similar to membrane targeting compounds. Maybe the authors can discuss about this in more detail.

We agree with the reviewer that linezolid displayed a mean *t_0_* value closer to the *t_0_* values recorded for antibiotics with a membrane target compared to the *t_0_* values measured for antibiotics with an intracellular target (e.g. *t_0_* = 437, 6614, and 364 s for linezolid, roxithromycin and polymyxin B, respectively). However, when considering the other five accumulation parameters linezolid displayed averaged values closer to the values measured for antibiotics with an intracellular target (e.g. *k_1_* = 1.6, 4.4, and 229 a.u. s^-2^, *F_max_* = 512, 1034, and 2264 a.u., *k_2_* = 0.0001, 0.0001, and 0.007 s, *d_r_* = 0.0003, 0.001, and 0.09 s^-1^, *d_c_* = 0.0006, 0.0003, and 0.01 s^-1^ for linezolid, roxithromycin, and polymyxin B, respectively).

We have now explicitly acknowledged this in the revised manuscript on lines 233-237.

It is tempting to speculate that linezolid diffuses via *E. coli* outer membrane porins faster than larger or more hydrophobic antibiotics (such as roxithormyin and ciprofloxacin, respectively) while being pumped out more efficiently via efflux pumps. This hypothesis might explain the measured kinetic parameters reported above. In our opinion, we should not formally advance this hypothesis in this manuscript without further experimental support. We believe that this hypothesis should instead be further investigated in a follow up study.

– Figure 2. I don't see the value of this figure. The data shown in panels A and B are quite scattered and the correlations are not so obvious.

Following the reviewer's suggestion we have now moved this figure to the Supplementary Information (new Figure 1—figure supplement 15) and summarised the paragraph describing these correlations on lines 263-281 of our revised manuscript. We have also added further information to this paragraph in response to comment 5 of reviewer 2, please see below.

– Data in Figure 3C and 5A are plotting correlations of the same two parameters, but the values shown are different.

The data in Figure 3C (Figure 2C in our revised manuscript) report single-cell elongation rates *during* treatment with roxithromycin-NBD whereas the data in Figure 5A (Figure 4A in our revised manuscript) report single-cell elongation rates *before* treatment with roxithromycin-NBD as detailed in the caption of each figure and discussed in the manuscript. Therefore, the two figures report two different datasets. These two datasets are also reproduced in Figure 4—figure supplement 2A to allow for comparisons between the elongation rate before and during treatment with roxithromycin-NBD as discussed on lines 439-441 of our revised manuscript. We have now also explicitly acknowledged that the data in Figure 4—figure supplement 2A are reproduced from Figure 2C and Figure 4A to further clarify this point.

– Ln 448-451 – The authors generated a tolC reporter strain to address the link between kinetic parameter t0 and tolC expression. However, they do not show this correlation but based on positive correlations between t0 and elongation rate and between GFP and elongation rate, conclude that it is in line with their hypothesis. This is erroneous. This would need comparing of GFP levels with t0 and correlating directly. Similar logic was used for in subsequent paragraphs for panels 5E and 5F.

We agree with the reviewer, we have now amended Figure 4 accordingly, please see our essential revision 3 above.

Reviewer #2 (Recommendations for the authors):– To check whether the dye has an effect on the drug activity, it would be informative to compare single cell growth variability with labeled and unlabeled compounds.

We agree with the reviewer and for this reason we did compare single-cell elongation rate both before and during treatment with both labelled and unlabelled roxithromycin, please see Figure S14 of the submitted manuscript (now Figure 4—figure supplement 2). We have now explicitly acknowledged that we found large single cell growth variability in the presence of both labelled and unlabelled roxithromycin (CV of 59 and 44%, respectively) in accordance with previous data obtained with unlabelled antibiotics (see for example Figures 2 and 3 in Baltekin et al*., PNAS* 114, 9170, 2017).

This information has been added on lines 446-447 of our revised manuscript.

– Background fluorescence is analyzed in empty channels. Another relevant control to rule out that autofluorescence affects the results is to measure fluorescence in cells that are treated with the non-fluorescent versions of the antibiotics.

We agree with the reviewer and we did measure autofluorescence of unlabelled roxithromycin and ciprofloxacin using the same excitation conditions used for their fluorescent derivatives (0.03 s exposure to the blue excitation band of a broad-spectrum LED operated at 8 mW). Under these conditions the drug autofluorescence detected from the bacteria was not distinguishable from the background (i.e. from empty channels). We have now explicitly acknowledged this on lines 197200 and 442-444 of our revised manuscript. However, please note that unlabelled ciprofloxacin was distinguishable from the background upon 0.1 s exposure to the UV excitation band of a broadspectrum LED operated at 40 mW, please see data reported in Figure S10A of the submitted manuscript (now Figure 1—figure supplement 12A). We have now explicitly acknowledged this on lines 200201 of our revised manuscript.

– Quite a few claims and implications seem exaggerated:– Line 569: the authors claim to mechanistically understand why some drugs are more effective in biofilms than others. Whilst their Figure 4 is interesting, and seems in line with the cited references, it remains unclear why some drugs show a trend as a function of position whereas others don't.

We agree with the reviewer that this statement might have been misleading. We have therefore, rephrased it as follows on lines 604-606:

"Our data are in agreement with previous work in clinical settings suggesting that macrolides, quinolones, and oxazolidinones are more effective within infecting biofilms compared to glycopeptides and polymyxins."

– Identifying the target of antibiotics via single cell drug uptake kinetic measurements seems a bit farfetched as an application. First, following the data provided by the authors, one could also classify the drugs using (more high throughput) bulk measurements. Second, the authors should compare their technique in the context of current approaches used to classify drugs to clarify its advantages and disadvantages. Lastly, in line 292 it says that this technique can help 'phenotyping bacterial populations'. What do the authors mean here? Microfluidic techniques to rapidly phenotype drug sensitivity have already been proposed without using fluorescent antibiotics – it is not clear how much fluorescence would add here.

We agree with reviewer and we have now rephrased this sentence as follows on lines 295-297 of our revised manuscript:

"Taken together, these data suggest the existence of a unique accumulation pattern for the specific antibiotic in use and could be used in combination with existing single-cell microfluidic platforms to rapidly phenotype drug sensitivity (Baltekin et al.*, PNAS* 114, 9170, 2017; Bakshi et al*., Nat Microbiol* 6, 783, 2021; Bergmiller et al.*, Science* 356, 311, 2017), ultimately in clinical antibiotic testing.”

– Line 491 suggests that reducing heterogeneity is the main goal of any treatment. However, the more established goal is to inhibit the growth of as many cells as possible. This can be achieved by keeping the same heterogeneity but simply reducing the mean.

We agree with the reviewer that this sentence might have been misleading and therefore we have removed it from our revised manuscript.

– In line 515 ‘needs to be’ seems exaggerated; similarly ‘major rethink’ in line 543.

We have now changed line 550-551 to ‘a finding that *could inform the design of antibiotic therapy using macrolides’* and line 579-580 to ‘This new knowledge *deepens our understanding* of phenotypic resistance to antibiotics.’

– The term ‘phenotypically engineer’ is used both in the abstract and in line 581. This term is unclear for adding an extra compound.

We have now changed the sentence in the abstract to ‘*We used this new knowledge to eradicate* variants that displayed low antibiotic accumulation’ and the sentence on line 616-617 to ‘We show that it is possible to eradicate phenotypic variants currently avoiding antibiotic accumulation clonal bacterial populations.’

– Figure 3a shows four exemplary curves out of a large data set. It would help to rigorously analyse the reduction in growth rate after the supposed lag time for the entire dataset.

We did compare the elongation rate before and after uptake for the entire dataset, please see Figure 3C and 3D (now Figure 2C and 2D) of the submitted manuscript. We have now explicitly stated on line 335 of our revised manuscript that the findings in Figure 2C and 2D were obtained by analysing the entire dataset.

– Line 587: The ‘biases’ mentioned here need to be explained. As is, this claim is unclear without digging into literature.

We agree with the reviewer that this last part of the sentence requires further explanations that go beyond the scope of the current manuscript. Therefore, we have rephrased the last sentence of the conclusions as follows on lines 619-622 of our revised manuscript: ‘Finally, our novel single-cell approach reveals that each antibiotic is characterised by a unique accumulation pattern and thus could in future be employed to simultaneously characterise the accumulation and efficacy of new leading antibiotic compounds.’

[Editors’ note: further revisions were suggested prior to acceptance, as described below.]

Reviewer #1 (Recommendations for the authors):The authors have addressed and made modifications to the revised text and have incorporated these new data in the revised manuscript.I am satisfied with the current version of the manuscript.Only one important edit. Figure 4D-F needs to be updated and reflect the new content being described in the text and figure legend. The correct figure was shown in the response to the reviewers letter but the figure in the main file is still the previous version.

We thank the reviewer for recommending our manuscript for publication, we have now updated Figure 4 in the manuscript file.

Reviewer #2 (Recommendations for the authors):The manuscript has certainly improved, especially due to the inclusion of new control experiments and some simulations. However, some of my concerns remain.1. I am not convinced that the intra-experiment heterogeneity in elongation rate is similar to that reported in literature as claimed in the response. For example, the elongation rates shown in Figure 2 in Wang et al., are approximately normally distributed with almost all data falling between 0.03/min and 0.06/min; no elongation rates are anywhere near zero. Similarly, in Baltekin et al., Figure 2F (left panel) virtually all (normalised) growth rates fall between 0.5 and 1.25 with a handful of lower values and no values near zero. In both papers, the numbers of cells analyzed are orders of magnitude larger, making outliers more likely. In the preset work, while fewer cells were analyzed, the elongation rates are spread from 0 to 15 μm/h with many values near zero (Figure 4). Therefore, I cannot follow the authors' reasoning here: the variability in elongation rates they observed is clearly much greater than what has been reported in literature. This is problematic for the reasons described in my first report.

We agree with the reviewer that the variability in the elongation rate before antibiotic treatment measured in our study was larger compared to that measured in previous reports (Wang et al*., Current Biology* 20:1099, 2010; Baltekin et al*., PNAS* 114:9170, 2017). This discrepancy is due to the fact that we started to measure single-cell elongation rates immediately upon loading an aliquot of stationary phase *E. coli* in the mother machine (and then averaged over all values obtained for each cell during the 2 h growth period). In contrast, previous studies either did not measure single-cell elongation rates of the first ten generations upon loading *E. coli* in the mother machine (Wang et al*., Current Biology* 20:1099, 2010) or pre-cultured *E. coli* for 2 h before loading in the mother machine and thus did not measure single-cell elongation rates during the 2 h regrowth period from stationary phase (Baltekin et al*., PNAS* 114:9170, 2017). Therefore, our elongation rate measurements capture heterogeneous growth resumption from stationary phase (Levin-Reisman et al*., Nature Methods* 7:737, 2010; Joers et al*., Scientific Reports* 6:24055, 2016), with a minority of bacteria that did not divide in the 2 h growth prior to antibiotic exposure (i.e. 10%, non dividing Figure 4—figure supplement 2A-C), a few bacteria that divided once (i.e. 30%, slow dividing subgroup, Figure 4—figure supplement 2D-F), and the majority of bacteria that divided two times (i.e. 60%, fast dividing subgroup, Figure 4—figure supplement 2G-I). Elongation rates within bacteria that divided once fell within 1.5 and 5 µm hr^-1^, elongation rates within bacteria that divided twice fell within 4 and 12 µm hr^-1^, with a within subgroup variation of 3-4 folds in line with previous reports (Wang et al*., Current Biology* 20:1099, 2010; Baltekin et al*., PNAS* 114:9170, 2017). Importantly, cells within both the slow and fast dividing subgroups displayed a significant correlation between elongation rate (before treatment) and *t_0_* or *F_max_* (Figure 4—figure supplement 2).

These data are discussed on page 16, lines 447-461 of the revised manuscript and in the new Figure 4—figure supplement 2.

Finally, we had already performed separate experiments flowing LB for a longer period of time and found that the elongation rate (averaged over the whole bacterial population) did not further increase after the first 2 h exposure to LB (Figure 1—figure supplement 1), as detailed in our previous response to Essential revision 1 and on lines 98-101 of the current manuscript. In fact, by the 2 h time point over 90% of the bacteria had reached an elongation rate in range 4-12 µm hr^-1^.

2. The authors argue that dilution via growth is unlikely to play a role because they rarely observed cell division during their experiment. However, the two fundamental timescales here are the doubling time and the drug uptake time. The authors state that the doubling time is 75 min = 4500 seconds, while uptake of roxithromycin takes more than 5000 seconds (Figure 1). Therefore, regardless of whether the experiment was performed long enough to observe cell division, drug uptake is slow compared to dilution via growth of cell volume (which happens continuously irrespective of cell divisions). This effect therefore inevitably contributes to the net drug uptake kinetics. One could try to quantitatively correct for this effect; this would be important to rule out that it largely explains the observed correlation between growth rate and net uptake.

We agree with the reviewer that growth in cell volume during roxithromycin treatment helps bacteria in diluting the intracellular drug concentration. In fact, we measured a 3-fold variation in elongation rate (and thus in cell volume considering that the other two cell dimensions are physically constrained in all cells due to the device geometry), with the slowest and fastest bacterium displaying an elongation rate of 0.7 and 2.3 µm hr^-1^, respectively, during treatment with roxithromycin at a growth inhibitory concentration (i.e. 192 µg ml^-1^, Figure 4—figure supplement 3). Therefore, the measured variation in roxithromycin accumulation is due in part to dilution of intracellular drug concentration via differential cell growth. To account for this effect, we had already integrated the rate constant *d_c_* in equation 1 of our phenomenological mathematical model. This parameter accounts for drug loss via efflux, degradation or dilution via growth (please see page 8, lines 213-216 and Methods, page 26). Moreover, other factors (including ribosome and efflux pump abundance) must also play a role in the measured cell-to-cell differences in roxithromycin accumulation. In fact, these differences include a 160-fold variation in *t_0_* (30 s < *t_0_* < 4900 s), a 60-fold variation in *k_1_* (0.2 a.u. s^-2^ < *k_1_* < 12 a.u. s^-2^) and a 12-fold variation in *F_max_* (250 a.u. < *F_max_* < 3000 a.u.) far larger than the measured variation in elongation rate.

These data are discussed on page 17, lines 482-494 of the revised manuscript and in the new Figure 4—figure supplement 3.